# BENCHMARKING VISUAL COGNITION OF MULTI-MODAL LLMS VIA MATRIX REASONING

## ABSTRACT

Recently, Multimodal Large Language Models (MLLMs) and Vision Language Models (VLMs) have shown great promise in language-guided perceptual tasks such as recognition, segmentation, and object detection. However, their effectiveness in addressing visual cognition problems that require high-level multi-image reasoning and visual working memory is not well-established. One such challenge is matrix reasoning – the cognitive ability to discern relationships among patterns in a set of images and extrapolate to predict subsequent patterns. This skill is crucial during the early neurodevelopmental stages of children. Inspired by the matrix reasoning tasks in Raven's Progressive Matrices (RPM) and Wechsler Intelligence Scale for Children (WISC), we propose a new dataset MaRs-VQA and a new benchmark VCog-Bench to evaluate the zero-shot visual cognition capability of MLLMs and compare their performance with existing human visual cognition investigation. Our comparative experiments with different open-source and closed-source MLLMs on the VCog-Bench revealed a gap between MLLMs and human intelligence, highlighting the visual cognitive limitations of current MLLMs. We believe that the public release of VCog-Bench, consisting of MaRs-VQA, and the inference pipeline will drive progress toward the next generation of MLLMs with human-like visual cognition abilities.

## 1 INTRODUCTION

Matrix reasoning is a crucial ability in human perception and cognition, essential for nonverbal, culture-reduced intelligence measurements as it can minimize the influence of acquired knowledge and skills (Jensen, 1998; Jaeggi et al., 2010; Laurence & Macedo, 2023). Common matrix reasoning problems consist of images with simple shapes governed by underlying abstract rules (Małkiński & Mańdziuk, 2023) (see Figure 1). Participants have to identify and comprehend the rules based on a few provided patterns, and then reason about the next pattern following the same rules. Matrix reasoning is an important reflection of many fundamental capabilities of human intelligence, such as processing speed and working memory, that emerge in the early stage of children's neurodevelopment (Gentner, 1977). To quantitatively measure human's intelligence using matrix reasoning, many assessment methods have been proposed as a part of fluid intelligence tests. The two most famous assessments are Wechsler Intelligence Scale for Children (WISC) (Wechsler & Kodama, 1949) and Raven's Progressive Matrices (RPM) (Raven, 2003).

In computer vision, matrix reasoning tasks have emerged as an ideal testbed for investigating whether deep learning models can match or even surpass human cognitive abilities, motivating the creation of diverse problem settings and datasets (Chollet, 2019; Małkiński & Mańdziuk, 2023; Barrett et al., 2018; Zhang et al., 2019; Webb et al., 2020). Previous research on matrix reasoning assessments applied typical machine learning settings – finetuning models on training sets and evaluating the performance on test sets (Hu et al., 2021; Małkiński & Mańdziuk, 2022; Zhao et al., 2024). However, in human psychometrics, matrix reasoning are designed to assess visual reasoning abilities without prior specific training on similar tasks, which is similar to the zero-shot learning problem, but not training-testing paradigm in machine learning. Children taking these tests typically do not receive any specialized training in matrix reasoning beforehand. Instead, they rely on their general cognitive skills developed through everyday experiences in natural scenes. Previous machine learning models ignore these prerequisites when modeling matrix reasoning problem. This could lead to an

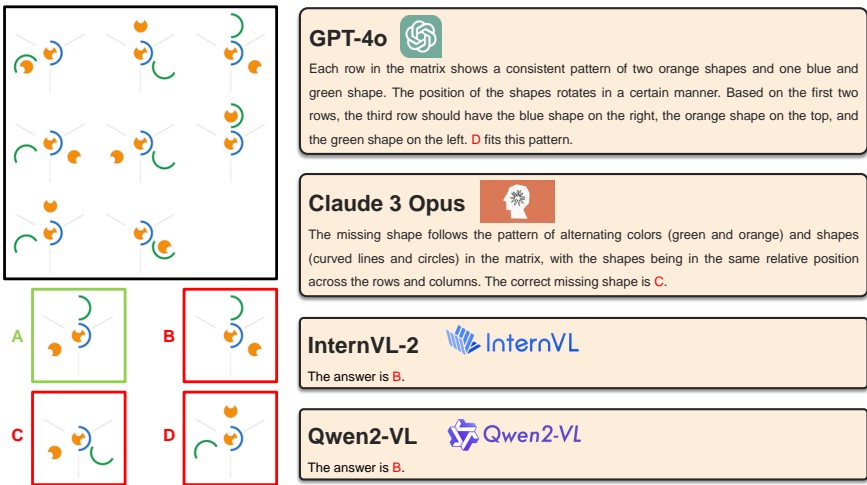

Figure 1: The example of the subpar performance of current state-of-the-art MLLMs (GPT-4o, Claude 3 Opus) and open-sourced VLMs (InternVL-2, Qwen2-VL) on a simple matrix reasoning task used in MaRs-VQA (similar to cases in RPM and WISC). Both models can recognize the basic shapes in the provided patterns but fail to reason the next pattern.

overestimation of the models' reasoning abilities, as they might be overfitting with training data rather than demonstrating genuine generalization and reasoning skills from visual cognition.

Recently, Multimodal Large Language Models (MLLMs) have shown surprising understanding and reasoning capabilities, marking an important milestone towards Artificial General Intelligence (AGI) (Chollet, 2019; Ji et al., 2022; Peng et al., 2023). These models are learned from a large amount of data in the general domain and are proven can be generalize to unfamiliar tasks without prior exposure by in-context learning. However, current MLLMs remain inadequate in visual cognition problems that require higher-level inductive reasoning (Yang et al., 2023). An example is their poor performance on the RAVEN IQ-test (Huang et al., 2024; Fu et al., 2024), which heavily relies on abstract reasoning skills. The RAVEN IQ-test also has some limitations, including a small dataset of only 50 samples (Huang et al., 2024), which may introduce randomness and fail to comprehensively and robustly evaluate MLLMs. Besides, it doesn't include a comparative study with human performance.

To address the matrix reasoning assessment and the deficiencies of existing cognitive testing benchmarks, we propose a new visual question answering (VQA) dataset – MaRs-VQA, which is the largest psychologist-verified VQA dataset for matrix reasoning assessment including 1,440 examples in total. The sample diversity of MaRs-VQA also surpasses other datasets before. It contains over 50 types of shape, 16 types of colour and over 500 graphic combinations. We also introduce VCog-Bench, the first zero-shot matrix reasoning benchmark to evaluate MLLMs' visual cognition. In VCog-Bench, We conduct thorough evaluation and comparison across 16 existing MLLMs (including their variants) and human performance under a zero-shot inference setting (no prior knowledge) on MaRs-VQA and other abstract reasoning datasets containing human studies. In our experiments, we observe that MLLMs with more parameters generally perform better on our benchmark, adhering to established scaling laws in a limited scope. However, even the largest open-source MLLMs and GPT-4o fall short of surpassing human performance in matrix reasoning tasks. Furthermore, many MLLMs have a mismatch in performance between matrix reasoning tasks and other general VQA benchmarks, which provides some insights into the drawbacks of existing models. In conclusion, our contributions are summarized as follows:

- We introduce a new matrix reasoning VQA dataset – MaRs-VQA, containing 1,440 image instances designed by psychologists, which is the largest dataset for matrix reasoning zero-shot evaluation.
- We propose VCog-Bench, the most comprehensive visual cognition benchmark to date, which evaluates the matrix reasoning performance of 16 existing MLLMs and comparing them with human's performance.

- Our thorough experiments qualitatively reveal the visual cognition gap between MLLMs and humans in matrix reasoning problems. We also show additional insights of deficiencies in MLLMs, which can inspire more future investigations in model design.

## 2 RELATED WORKS

| Dataset | Source | Sample | Instance | RGB image | Human Study | Psychological Validity | Open-source | VQA Annotation |
|---|---|---|---|---|---|---|---|---|
| kosmos-iq50 (NeurIPS-23) (Huang et al., 2024) | RAVEN-IQ Test | | 50 | ✗ | ✗ | ✓ | ✗ | ✗ |
| Visual Reasoning Benchmark (COLM-24) (Zhang et al., 2024c) | Mensa Test, RAVEN, IntelligenceTest | | 241 | ✗ | ✗ | ✗ | ✗ | ✗ |
| MaRs-VQA (ours) | MaRs-IB | | 1,440 | ✓ | ✓ | ✓ | ✓ | ✓ |

Table 1: Comparison of recently released zero-shot matrix reasoning datasets to evaluate MLLMs.

**Cognitive Test of Large Language Models (LLMs)** The rise of LLMs has aroused interest in exploring human-like AI in psychology and cognition (Ullman, 2023). Recent works tested LLMs' cognitive abilities in causal reasoning (Binz & Schulz, 2023), abstract reasoning (Xu et al., 2023b; Moskvichev et al., 2023; Jiang et al., 2024b; Ahrabian et al., 2024), analogical reasoning (Webb et al., 2023), systematic reasoning (Hagendorff et al., 2023), and theory of mind (Strachan et al., 2024). Their observation showed that LLMs like GPT-4 (Achiam et al., 2023) have been proven successful in most cognitive tests related to language-based reasoning. Despite this success, only limited research has been conducted on the areas of MLLMs and visual cognition. Visual cognition involves the process by which the human visual system interprets and makes inferences about a visual scene using partial information. *Buschoff et al.* observed that while LLMs demonstrate a basic understanding of physical laws and causal relationships, they lack deeper insights into intuitive human preferences and reasoning. Almost all existing visual cognition benchmarks focus on testing MLLMs' cognitive abilities in simple tasks (Lerer et al., 2016; Zhou et al., 2023; Jassim et al., 2023), and ignore testing complex abstract reasoning and logical reasoning ability related to fluid intelligence. Therefore, new and challenging benchmarks based on the theory of visual cognition are needed to assess and improve AI systems' capabilities for human-like visual understanding.

**Matrix Reasoning** Matrix reasoning is often used to determine human intelligence related to visual cognition and working memory (Salthouse, 1993; Jaeggi et al., 2010; Fleuret et al., 2011) that is widely used by RPM (Raven, 2003; Soulières et al., 2009), WISC (Wechsler & Kodama, 1949; Kaufman et al., 2015) to evaluate human's ability to detect the underlying conceptual relationship among visual objects and use reasoning to find visual cues. Early research indicated that deep learning models can be trained with large-scale matrix reasoning datasets to solve simple matrix reasoning (Stabinger et al., 2021; Małkiński & Mańdziuk, 2022; 2023; Xu et al., 2023a; Małkiński & Mańdziuk, 2024) and compositional visual relation tasks (Fleuret et al., 2011; Zerroug et al., 2022; Ommer & Buhmann, 2007; Liu et al., 2021), achieving human-level accuracy. Several datasets and benchmarks are also proposed, such as PGM (Barrett et al., 2018), RAVEN (Zhang et al., 2019), RAVEN-I (Hu et al., 2021), RAVEN-FAIR (Benny et al., 2021), CVR (Zerroug et al., 2022). However, these works have a key limitation. They ignore that humans can solve these problems by zero-shot reasoning without explicitly learning from large-scale data. After the blooming of LLMs, researchers are keen on testing whether LLMs reached the same abstract reasoning capabilities as humans. *Webb et al.* (Webb et al., 2023) encode matrix reasoning into a symbolic problem based on human's prior and validate LLM can understand this task. Recently, there are also some useful zero-shot visual reasoning inference datasets containing matrix reasoning samples have been proposed in the AI/ML community, such as RAVEN-IQ (Huang et al., 2024) containing 50 instances, Visual Reasoning Benchmark (Zhang et al., 2024c) containing 241 instances in total, but all of them are limited by

lacking rigorous human experiments as reference and conducting experiments on relatively small datasets without psychometrical validation.

**Vision-Language Models** Researchers have been actively investigating the utility of Vision-Language Models (VLMs) for addressing vision reasoning tasks (Zellers et al., 2019; Bordes et al., 2024). These latest VLMs are constructed using a combination of the CLIP vision encoder, pre-trained LLMs, and a connected adapter to align visual features with language space (Zhang et al., 2024b; Shao et al., 2024; Gupta & Kembhavi, 2023; Fu et al., 2024). Notably, methodologies such as MiniGPT-4 (Zhu et al., 2023), InstructBLIP (Dai et al., 2024), LLaVA (Liu et al., 2024b), CogVLM (Wang et al., 2023) underscore the significance of employing high-quality visual instruction tuning data. Additionally, tool learning methods have also explored the potential of integrating code generation pipelines with visual inference (Surís et al., 2023). Nevertheless, current VLMs encounter challenges in adapting to high-resolution and visually complex images. These problems stem from the absence of a robust visual search mechanism (Wu & Xie, 2023), few-shot reasoning (Guo et al., 2023), compositional understanding (Yuksekgonul et al., 2022) and the constrained visual grounding capabilities inherent in CLIP (Tong et al., 2024).

## 3 MaRs-VQA Dataset

The MaRs-VQA dataset is designed to evaluate the zero-shot abstract reasoning capabilities of MLLMs through various matrix reasoning VQA tasks. The images in MaRs-VQA are sourced from the questionnaires in Matrix Reasoning Item Bank (MaRs-IB) (Chierchia et al., 2019), which is created by psychologists including 18 sets of abstract reasoning questionnaires (80 instances in each set) for non-verbal abstract reasoning assessment of adolescents and adults. Each item presents an incomplete $3 \times 3$ matrix of abstract shapes, requiring participants to identify relationships among the shapes. Then, we create VQA annotations in the images from all questionnaires.

To transform the matrix reasoning problem into a VQA task, we firstly define three different option sets – two image-based sets (A and B) and one language-based set (C). In Option Set A, we provide four candidates to the missing patch in the question. In Option Set B, the options are created by filling the four patches in Set A into the $3 \times 3$ question image. Note that Option Set B is used for visualization purposes only and is not included in our experiment. We further diversify the modalities of our dataset to support the evaluation of different kinds of models. Specifically, we use GPT-4o and human annotators to generate language-based descriptions for each option, forming Option Set C. In the data generation process, we first manually design 10 VQA examples, which serve as the initial human annotations in our data collection. These examples are then used as few-shot samples to query GPT-4o through in-context learning. The context generation system prompt guides GPT-4o to compare all four option images and generate distinct descriptions for each one. After generating all samples, human annotators in the author team review each option and revise the incorrect description. Examples are showed in Figure 6 in the Appendix. In Table 1, Compared with other matrix reasoning datasets for MLLM's visual cognition evaluation, MaRs-VQA is the largest one with unique features on psychological validity, human study reference, VQA annotations.

## 4 Visual Cognition Benchmark (VCog-Bench)

Different from the training-testing paradigm setting in other abstract visual reasoning datasets like RAVEN (Zhang et al., 2019), our goal of MLLM agent in VCog-Bench is to complete the $3 \times 3$ matrix by finding the missing cell from multiple options by **zero-shot learning** under the same setting in human's matrix reasoning test. To this end, MLLM agents have to deduce relationships across the other cells of the matrix and infer the missing cell accordingly. Based on the current progress of Multimodal LLMs, we propose two potential solutions as baselines for VCog-Bench.

### 4.1 Multi-Image Reasoning via Chain-of-Thought (CoT)

Recent research in the NLP community has revealed the effectiveness of CoT in improving the reasoning capability of LLMs for complex problems (Wei et al., 2022; Kojima et al., 2022). In this paper, we propose the object-centric CoT prompting strategy, which combines the ideas of CoT (Zhang et al., 2023; Zhou et al., 2024; Zhang et al., 2024a), object-centric relational abstraction (Webb et al.,

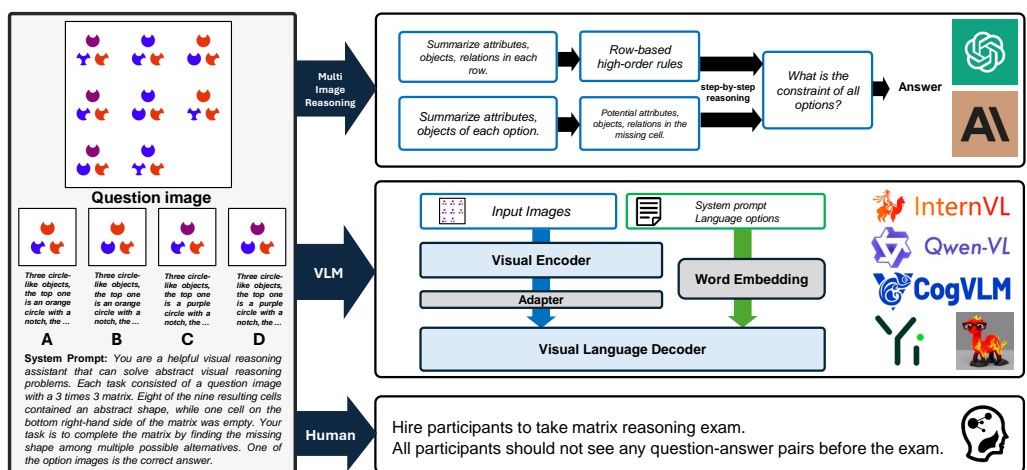

Figure 2: An overview of the VCog-Bench. The left part is the model input, including a question image, multiple option images and a system prompt describing the task. The right part shows the step-by-step CoT for multi-image reasoning and VLM solution for matrix reasoning problems.

2024a;b; Mondal et al., 2024; Xu et al., 2023b) and object-centric representation learning (Seitzer et al., 2022; Dittadi et al., 2022; Jiang et al., 2024a), to enhance the MLLM's zero-shot learning performance in solving matrix reasoning problems.

Following previous works (Carpenter et al., 1990; Barrett et al., 2018; Chierchia et al., 2019; Zhang et al., 2019), we formulate the structure $K$ of matrix reasoning as a combination of four components, $K = \{[r, a, o, s] | r \in \mathcal{R}, a \in \mathcal{A}, o \in \mathcal{O}, s \in \mathcal{S}\}$. $\mathcal{R}$ is a set of rules of how the pattern changes along each row and column (*e.g.*, rotating by a fixed angle and shifting by a fixed distance); $\mathcal{A}$ is a set of attributes in each pattern (*e.g.*, color, shape, and size); $\mathcal{O}$ is how to integrate objects in each cell (*e.g.*, spatial location and overlap); $\mathcal{S}$ denotes a set of constraints for designing answer options (*e.g.*, options should have minimum difference), which avoids that participants solving the matrix reasoning problems in unintended ways.

Based on structure $K$, we use three stages to guide MLLM to use human-level thought to understand matrix reasoning tasks. The first stage is to guide the Multimodal LLM to summarize the visual feature (e.g. shape) of each row in the $3 \times 3$ question image. Then, based on these row-based visual features, the model will then conclude the high-order rule/pattern $\mathcal{R}$. The second stage is to extract the basic attributes $\mathcal{A}$ and inner relations $\mathcal{O}$ to integrate objects in each option image. The third stage is to infer the answer based on exclusion with potential answer designed constraints $\mathcal{S}$. The system prompt of CoT will guide MLLM to step-by-step infer the sub-conclusion of each stage. And finally give the answer. The Multi-Image Reasoning section of Figure 2 shows a schematic depiction of how to leverage CoT in matrix reasoning tasks.

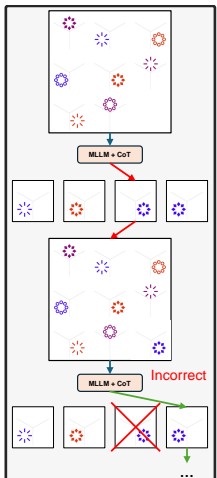

Figure 3: Multi-round CoT.

To further enhance CoT with diverse prompts, we introduce a multi-round architecture (Figure 3) inspired by the Monte Carlo Tree Search from Tree-of-Thought (ToT)(Yao et al., 2024). In the first reasoning round, the MLLM apply multi-image CoT solve the matrix reasoning problem. The selected image is then incorporated into the question image as a new input, which is fed back into the MLLM with a prompt directing it to evaluate the correctness of the complete $3 \times 3$ matrix, specifically focusing on the bottom-right corner. If the MLLM determines the result is correct, the final answer is output; otherwise, the incorrect option is excluded and CoT process is repeated.

## 4.2 VISION-LANGUAGE MODELS (VLMs)

In addition to MLLMs, we also evaluate the performance of VLMs for a thorough comparison. In VLMs, we only use question image as visual input and transform all option images into language

descriptions (*i.e.*, Option Set C), which matches the input representations required by VLMs (Xu et al., 2023b; Camposampiero et al., 2023). The VLM section in Figure 2 illustrates this pipeline.

The test set contains $n$ VQA samples, denoted as $\{(\mathbf{q}_i, \mathbf{x}_i, \mathbf{y}_i)\}_{i=1}^n$. $\mathbf{q}_i$ represents the question image showing the $3 \times 3$ matrix reasoning task (MaRs-VQA). $\mathbf{x}_i = [x_i^1, ..., x_i^k]$ represents the context description in the option set, where $k$ is the number of options. $\mathbf{y}_i$ is the answer of the matrix reasoning question. The zero-shot inference pipeline of VLM can be formulated as:

$$\hat{\mathbf{y}}_i = F_\theta(\mathbf{q}_i, \mathbf{x}_i, \mathbf{x}_{sys}). \tag{1}$$

$\mathbf{x}_{sys}$ is the system prompt, including independent information about the matrix reasoning problem setting, structure $K$ for each dataset and requirements for the output format. $\hat{\mathbf{y}}_i$ is the prediction result .$F_\theta$ is an autoregressive decoder in the LLM for answer generation. It is defined as:

$$P(\hat{\mathbf{y}}_i|\mathbf{q}_i, \mathbf{x}_i, \mathbf{x}_{sys}) = \prod_{j=1}^{L} P(\hat{\mathbf{y}}_{i,j}|f(\mathbf{q}_i), \mathbf{x}_i, \mathbf{x}_{sys}, \hat{\mathbf{y}}_{i,<j}; \theta), \tag{2}$$

where $f$ is the visual encoder and adapter layer, $L$ is the sequence length of answers and $\hat{\mathbf{y}}_{i,<j}$ is all answer tokens before $\hat{\mathbf{y}}_{i,j}$.

In VLMs, the input question image is first processed by the visual encoder such as CLIP (Radford et al., 2021). Then, additional adapter layers are used to map visual features into language feature space. These features, along with the context-based option descriptions, are sent to the LLM decoder. The LLM decoder then integrates the information from both the input question image and the option descriptions to address the VQA task. VLMs leverage the strengths of both visual encoders and language models, allowing for a more comprehensive analysis of the matrix reasoning problems. It provides a structured way to break down the problem, potentially improving interpretability compared to end-to-end close-source models.

## 5 EXPERIMENTS

### 5.1 EXPERIMENTAL SETTINGS

**Datasets** In addition to MaRs-VQA, we selected two well-known open-source datasets for matrix reasoning and abstract visual reasoning to conduct experiments in VCog-Bench. The first dataset is RAVEN (Zhang et al., 2019), designed to probe abstract reasoning in a format similar to the Raven's Progressive Matrices IQ test, with each question providing eight options. The second dataset is Compositional Visual Reasoning (CVR) (Zerroug et al., 2022), which evaluates deep learning models using 103 unique configurations generated by predefined rules. Each sample in CVR is an outlier detection problem, with four options provided per question. However, both RAVEN and CVR share a significant limitation: all samples are algorithmically generated using fixed rules, which limits their diversity and lacks psychological validity.

**Baselines for Multi-image Reasoning** We selected the Claude 3 family (Haiku, Sonnet, Opus) (Anthropic, 2024), GPT-4V (OpenAI, 2023), GPT-4o (OpenAI, 2024) as the primary multi-image CoT baselines as they support multiple images input and can generate reasoning process. The inputs for this task are all images, a question and multiple option images in Option Set A of Figure 6. Other open-sourced models are not included because they perform much worse than Claude and GPT and can not generate reasoning steps for matrix reasoning tasks.

**Baselines for VLMs** For the VLMs, we select state-of-the-arts open-source and closed-source models such as InstructBLIP (Dai et al., 2024), MiniGPT-v2 (Zhu et al., 2023), LLaVA-v1.6 (LLaVA-NeXT) (Liu et al., 2024a), CogVLMv2 (Wang et al., 2023), Yi-VL (Young et al., 2024), Qwen-VL (Bai et al., 2023), InternVL (Chen et al., 2024), Gemini Pro 1.5 (Reid et al., 2024), Claude 3 family (Haiku, Sonnet, Opus) (Anthropic, 2024), GPT-4V (OpenAI, 2023), GPT-4o (OpenAI, 2024) as the primary VLM baselines. The input is a question image and language-based options.

**Human Baseline** The human study results in Table 2 and 3 are reported from previous experiment results. The human subjects of RAVEN (Zhang et al., 2019) consists of college students from a

| Method | Learning | Accuracy (%) ↑ | | |
|---|---|---|---|---|
| | | MaRs-VQA (4-options) | RAVEN (8-options) | CVR (4-options) |
| Claude 3 Sonnet (Anthropic, 2024) | zero-shot | 22.92 | 10.71 | 27.83 |
| | CoT | 23.22 | 13.39 | 28.48 |
| Claude 3 Opus (Anthropic, 2024) | zero-shot | 20.85 | 11.61 | 26.86 |
| | CoT | 24.13 | 11.95 | 27.18 |
| Claude 3.5 Sonnet (Anthropic, 2024) | zero-shot | 23.18 | 14.08 | 25.97 |
| | CoT | 24.28 | 15.36 | 27.88 |
| GPT-4V (OpenAI, 2023) | zero-shot | 27.71 | 13.84 | 36.25 |
| | CoT | 33.13 | 15.63 | 40.62 |
| GPT-4o (OpenAI, 2024) | zero-shot | 30.21 | 19.20 | 42.50 |
| | CoT | 33.96 | 25.89 | 44.01 |
| Human | - | **69.15** | **84.41** | **78.70** |

Table 2: Experiments on multi-image reasoning. zero-shot means only provide the model system prompt about the matrix reasoning task definition. Chain-of-thought denotes the implementation in section 4.1. The results are averaged over three runs with three different random seeds.

| Method | Training Data | Model Scale | LLM Backbone | Accuracy (%) ↑ | |
|---|---|---|---|---|---|
| | | | | MaRs-VQA (4 Options) | RAVEN (8 Options) |
| InstructBLIP (Dai et al., 2024) | 129M | 7B | Vicuna-7B (Chiang et al., 2023) | 10.63 | 12.05 |
| LLaVA-v1.6 (Liu et al., 2024b) | 1.3M | 7B | Mistral-7B (Jiang et al., 2023) | 16.88 | 14.29 |
| MiniGPT-v2 (Zhu et al., 2023) | - | 8B | Llama-2-7B (Touvron et al., 2023) | 26.45 | 13.39 |
| Qwen-VL (Bai et al., 2023) | 1.4B | 10B | Qwen-7B (Bai et al., 2023) | 29.58 | 16.07 |
| InstructBLIP (Dai et al., 2024) | 129M | 13B | Vicuna-13B (Chiang et al., 2023) | 10.42 | 14.46 |
| CogVLMv2 (Wang et al., 2023) | 1.5B | 19B | Llama-3-8B (Meta, 2024a) | 26.46 | 12.05 |
| InternVL 1.5 (Chen et al., 2024) | 6.0B | 26B | InternLM2-Chat-20B (Cai et al., 2024) | 22.09 | 14.73 |
| Yi-VL (Young et al., 2024) | 100M | 34B | Yi-34B-Chat (Young et al., 2024) | 25.21 | 19.64 |
| LLaVA-v1.6 (Liu et al., 2024b) | 1.3M | 35B | Hermes-Yi-34B (Young et al., 2024) | 34.38 | 33.93 |
| InternVL 1.2+ (Chen et al., 2024) | 6.0B | 40B | Hermes-Yi-34B (Young et al., 2024) | 32.71 | 33.04 |
| Qwen2-VL (Wang et al., 2024) | - | 72B | Qwen2-72B (Yang et al., 2024) | 34.22 | 36.15 |
| InternVL 2 (Chen et al., 2024) | - | 76B | Hermes-2-Theta-Llama-3-70B (Teknium et al.) | 34.63 | 38.01 |
| Llama 3.2 (Meta, 2024b) | 6.0B | 90B | - | 34.81 | 35.26 |
| Claude 3.5 Sonnet (Anthropic, 2024) | unknown | unknown | unknown | 34.82 | 35.36 |
| GPT-4o (OpenAI, 2024) | unknown | unknown | unknown | 37.38 | 38.84 |
| Gemini Pro 1.5 (Reid et al., 2024) | unknown | unknown | unknown | 34.79 | 42.86 |
| Human | - | - | - | **69.15** | **84.41** |

Table 3: Experiments on using a question image and language descriptions for options as inputs to compare different VLMs. The results are averaged over three random seeds.

subject pool maintained by the Department of Psychology. Only "easily perceptible" examples were used in the investigation. CVR (Zerroug et al., 2022) hired 21 participants and each participant completed 6 different tasks with 20 problem samples for each task. The human study results of MaRs-IB (Chierchia et al., 2019) (data source of MaRs-VQA) are more rigorous. They are from 4 age groups ($N = 659$, aged 11–33 years). The accuracy for younger adolescents, mid-adolescents, older adolescents, and adults solving matrix reasoning in MaRs-IB are 61%, 68%, 73%, 81%. We use the average result of all groups in Table 2 and 3.

**Implementation** For closed-source baseline models, we establish basic prompts to introduce the matrix reasoning problem setting, which serve as the system prompt for zero-shot inference. For object-centric CoT reasoning, we create specific prompts to guide the model's thought process through multiple stages, enabling step-by-step reasoning. For open-source baseline models, we use the same system prompt settings across all models. Testing is conducted using two NVIDIA RTX 4090 GPUs for 7B-sized VLMs and eight NVIDIA A100 80GB GPUs for VLMs larger than 7B. All experiments are run with three different random seeds, and the results are averaged. We evaluate the results based on the accuracy of single-option matrix reasoning problems (Acc = Correct/Total), consistent with other VQA benchmarks (Lu et al., 2022; Liu et al., 2023).

## 5.2 EXPERIMENTAL RESULTS

In this subsection, we present the experimental results of the baselines in the VCog-Bench. The results demonstrate that while parts of baseline models can understand some basic forms of the matrix reasoning task, they struggle with complex tasks requiring both visual working memory and multi-image reasoning capability.

We divided our experiments into two parts. The first part involves end-to-end multi-image reasoning. For this experiment, we used multiple images as the input, including a question image and several option images (refer to Option Set A in Figure 6), and guided the MLLMs to decompose the problem into predefined structures before generating answers based on all available information. We tested

| Method | Multi-Image | Accuracy (%) ↑ | | | | |
|---|---|---|---|---|---|---|
| | | Level 1 (90) | Level 2 (96) | Level 3 (84) | Level 4 (72) | Level >4 (138) |
| Claude 3 Opus (Anthropic, 2024) | ✓ | 19.15 | 28.57 | 13.34 | 13.16 | 24.66 |
| GPT-4o (OpenAI, 2024) | ✓ | 57.78 | 27.08 | 27.38 | 19.43 | 21.74 |
| Claude 3 Opus (Anthropic, 2024) | ✗ | 24.44 | 25.00 | 40.48 | 38.89 | 39.13 |
| Gemini Pro 1.5 (Reid et al., 2024) | ✗ | 51.10 | 30.21 | 26.19 | 29.17 | 35.51 |
| GPT-4o (OpenAI, 2024) | ✗ | 58.89 | 45.83 | 32.14 | 26.39 | 26.09 |

Table 4: Compare closed-source MLLMs with different difficulty levels in MaRs-VQA. The number in the "()" is the number of case sample of selected level. The difficulty level is based on the complexity of color, size, geometry, positional relationships, and object counting.

the Claude 3 family, GPT-4V, and GPT-4o for this task, as these models can generate step-by-step multi-image reasoning. Table 2 shows that even the state-of-the-art closed-source MLLMs perform worse than humans in all matrix reasoning tasks. While object-centric CoT can help larger models achieve better performance, it does not benefit smaller models such as Claude 3 Sonnet. Compared to the results in MaRs-VQA and RAVEN, GPT-4o achieves much better zero-shot and object-centric CoT inference results in the CVR dataset, almost matching the performance (ResNet-50: 57.9%, ViT-small: 32.7%, WReN: 42.4%) of fine-tuned models with 1,000 training samples in CVR's paper (Zerroug et al., 2022).

In the second part of our experiment, we investigated the use of VLMs (question image + language options) to solve matrix reasoning problems in MaRs-VQA and RAVEN. The CVR dataset was excluded because the shapes it contains are too complex to describe accurately. As shown in Table 3, large-scale VLMs, such as Qwen2-72B and InternVL-2-76B, achieved comparable results to GPT-4o in MaRs-VQA and RAVEN. Notably, Gemini Pro 1.5 outperformed GPT-4o on the RAVEN dataset.

We identified three major issues after reviewing the reasoning outputs of current MLLMs in Table 2 and 3: (1) Limited Use of Visual Information: MLLMs cannot directly use visual features for reasoning, making them insensitive to non-verbal spatial features during CoT reasoning. This limitation is particularly evident when handling images that require describing the positional relations of objects. For example, it is difficult for MLLMs to distinguish each option in Figure 1 using language alone. (2) Restricted Visual Working Memory: The visual working memory of MLLMs is limited, causing visual feature information to be easily lost during the text generation reasoning process. (3) Integration Challenges: Even if MLLMs possess strong task-specific skills like recognition, segmentation, and object detection, they struggle to integrate these skills into high-level visual reasoning tasks. We will further analyse them in the ablation study.

## 5.3 ABLATION STUDY

In this subsection, we conduct ablation experiments to analyze how to improve the performance of MLLMs on the matrix reasoning problem. Table 5 compares the Chain-of-Thought (CoT) baseline with two approaches: few-shot reasoning and multi-round reasoning. Few-shot reasoning involves providing a small number of question-answer examples alongside the CoT system prompt. Multi-round reasoning employs the advanced CoT strategy illustrated in Figure 3. The results show that incorporating 1-shot and 3-shot question-option-answer pairs gradually increases the accuracy on MaRs-VQA from 34% to 36%. However, extending the number of examples to 5 does not yield further improvement. These findings suggest that while few-shot in-context learning helps the model better understand the matrix reasoning problem, it does not significantly enhance the MLLM's visual reasoning capabilities for these tasks. Additionally, using a multi-round tree search improves accuracy from approximately 34% to 42%, but it is considerably slower than single-round CoT, with each inference taking over 30 seconds in multi-round mode. We also compare different MLLMs across difficulty levels and different visual complexities in the MaRs-VQA dataset (see Table 4 and Table 6). The difficulty level in our tasks is defined by the number of sub-tasks (visual complexities) involved—specifically, variations in color, size, geometry (shape), positional relationships, and the presence of multiple objects. The results indicate that GPT-4o exhibits difficulty sensitivity similar to that of humans, whereas Claude 3 Opus does not demonstrate this ability. This is because GPT-4o can solve object size sub-tasks well in the MaRs-VQA, but is still struggling with other sub-tasks, especially the multi-object sub-task. More details are presented in the Appendix.

| Strategy | Accuracy (%) ↑ |
|---|---|
| CoT | 33.96 |
| CoT + 1-shot | 35.22 |
| CoT + 3-shot | 36.10 |
| CoT + 5-shot | 36.03 |
| multi-round tree search CoT | 41.96 |
| multi-round tree search CoT + 1-shot | 42.08 |

Table 5: Ablation on prompt selection.

| Visual complexity | Proportion (%) | Accuracy (%) ↑ |
|---|---|---|
| Shape | 68 | 33.96 |
| Color | 73 | 35.72 |
| Size | 16 | 63.26 |
| Position | 41 | 31.70 |
| Multi-Object | 71 | 31.48 |
| All | - | 33.96 |

Table 6: Ablation on visual complexity.

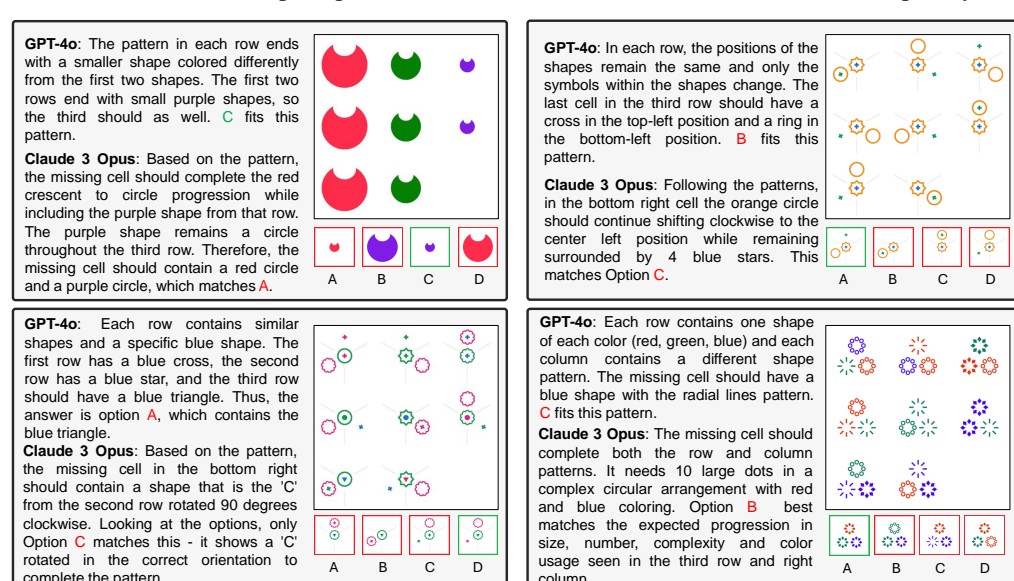

Figure 4: Different matrix reasoning problem (difficulty levels) from MaRs-VQA and MLLM's reply. We use green to represent correct answer and red to represent wrong answer of each question. The top left is a sample with difficulty level 1. The others are samples with difficulty level ≥4. The reasoning is a short summary of the CoT output, not the full version

## 5.4 QUALITATIVE ANALYSIS

In this subsection, we use case studies from the MaRs-VQA dataset to illustrate how MLLMs fail in some tasks and provide insights on how to improve MLLMs and VLMs for this task.

First, we present an example to explain why the Claude 3 family performs worse than GPT-4o and even worse than random guessing in most of experiments. Figure 4 top left is one of the most simple cases in MaRs-VQA's level 1 difficulty, Claude 3 Opus incorrectly identifies the shape as the main target of this matrix, while the actual target is the size. In contrast, GPT-4o correctly discerns the relationship between rows, noting: "The pattern in each row ends with a smaller shape colored differently from the first two shapes." This example highlights a critical shortcoming in Claude 3 Opus's reasoning ability: limited use of Visual information, demonstrating its struggle to accurately interpret the key attributes in matrix reasoning tasks. GPT-4o, on the other hand, showcases a superior understanding of the relationships within simple data, leading to more accurate responses.

However, the difficulty of the tasks increases, the performance of MLLMs deteriorates in multi-image reasoning. Figure 4 bottom left and shows an example, it is the level 6 difficulty containing shape, positional relation, shape with different objects. For these questions containing complex visual features, MLLMs tend to extract only a small portion of the key information from the question image. This limited extraction means that critical features are either overlooked or not effectively utilized in selecting the correct option. Consequently, the final answers are often incorrect or only partially related to the relevant attributes. It suggests that MLLMs are affected by the cognitive load associated with processing multiple sub-tasks simultaneously, which is closely related to the concept of visual working memory. The right two examples of Figure 4 also present the same observation. Additionally, we observed that GPT-4o is not sensitive to the positional relationships for multi-objects in the question images.

These failures highlight significant limitations in MLLM's visual processing capabilities. The model's inability to effectively leverage visual features and its lack of visual working memory result in incorrect interpretations. Furthermore, its insensitivity to positional relationships among multi-objects underscores a critical area for improvement in understanding and analyzing spatial information in visual reasoning.

## 5.5 VISUALIZATION

We also analyze the relationship between matrix reasoning accuracy and model scale in Figure 5. The figure illustrates the significant gap between MLLM's matrix reasoning performance and that of humans. This gap is substantial and suggests that simply increasing model size according to scaling laws will not be sufficient to bridge it.

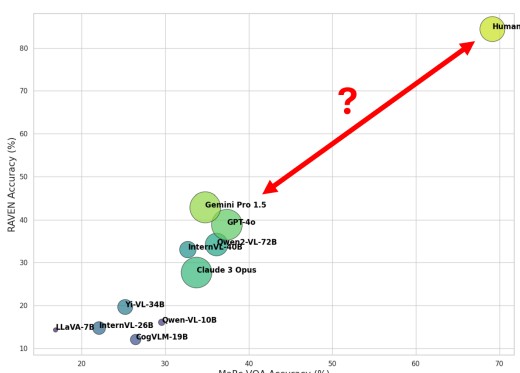

Figure 5: There is still a big gap between human's matrix reasoning capability and MLLM's. Bubble size corresponds to the model size. As we don't know the exact size of closed-source MLLMs, we set all of them to the largest value by default. The model size of human refers to the number of neurons (86B) in human's brain (Voytek, 2013).

## 6 DISCUSSION

**Social Impacts**  In the present work, we emphasize that zero-shot matrix reasoning is a key item to validate human-level intelligence, though it is still unclear how matrix reasoning ability is acquired early in human neurodevelopment. Children's visual reasoners (without any additional training) can provide sensible answers to matrix reasoning questions as early as age four. The long-term goal of our work is twofold. The first one is to explore the problem of how close AIs or MLLMs are to human-like cognitive abilities, which is raised by *François Chollet* in 2019 (Chollet, 2019). The second one is to develop an MLLM-powered AI agent that can simulate human-level zero-shot matrix reasoning capability. The agent will eventually guide vision generation models to generate new matrix reasoning samples and tasks and design new neurodevelopmental assessment tools. This will help psychologists and pediatricians explore and deconstruct how children activate such abilities in the early stage of neurodevelopment.

**Limitations**  An open-ended question is whether MLLMs need to achieve or surpass human-level zero-shot inference capability in matrix reasoning tasks. Addressing this issue requires new theories from cognitive science and psychology to accurately evaluate and compare human and MLLM intelligence. Unlike MLLMs, which rely on training data and domain-specific skills, human cognition develops gradually and evolves with age. Humans can also learn how to solve the problem progressively from previous seen matrix reasoning tasks while they are taking the test, but MLLM can not learn from it via in-context learning due to the maximum tokens length. Therefore, AI researchers, psychologists, and cognitive scientists must collaborate to rethink how to benchmark MLLM intelligence with human intelligence.

## 7 CONCLUSION

We introduce VCog-Bench, a publicly available zero-shot matrix reasoning benchmark designed to evaluate the visual cognition capability and intelligence of Multimodal Large Language Models (MLLMs). This benchmark integrates two well-known datasets RAVEN and CVR from the AI community and includes our newly proposed MaRs-VQA dataset. We also introduce several important concepts to redefine zero-shot matrix reasoning task evaluation, focusing on multi-image reasoning with object-centric Chain-of-Thought (CoT) system prompts. Our findings show that current state-of-the-art MLLMs and Vision-Language Models (VLMs), such as GPT-4o and LLaVA-1.6, InternVL demonstrate some basic understanding of matrix reasoning tasks. However, these models still face big challenges with complex situations and perform much worse than human. This highlights the need for further exploration and development in this area. By providing a robust benchmark, we aim to encourage further innovation and progress in the field of improving the visual cognition of MLLMs.

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

# Appendices

## CONTENTS

## A    DATASETS & BENCHMARKING CODE

We release the data and annotations of MaRs-VQA anonymously:

huggingface.co/datasets/vcog/marsvqa

We also release the initial version of code for MLLM inference in an anonymous github repo:

anonymous.4open.science/r/VCog-Bench-94D2

## B    DATA COLLECTION AND LICENSES

We showed and compared all datasets in VCog-Bench in Table 7. The data collection of VCog-Bench follows strict procedures. The reason we choose RAVEN, CVR, MaRs-VQA is because all these datasets contain zero-shot / few-shot human investigation results. Based on these results, we can compare the MLLM's performance with human in matrix reasoning tasks.

For RAVEN and CVR, we followed the original data generation pipeline in their repo. For MaRs-VQA, we download all questionnaires from MaRs-IB and then re-annotate all images by ourselves.

**RAVEN**    The original dataset link of RAVEN is github.com/WellyZhang/RAVEN. It is under GPL-3.0 License (RAVEN LICENSE) and is free to use by public. All data in RAVEN are generated by rule-based scripts. We follow the basic setting of RAVEN, and modify the range of COLOR_VALUES to $[255, 192, 128, 64, 0]$ and SIZE_VALUES to $[0.3, 0.45, 0.6, 0.75, 0.9]$. The sample size of RAVEN in VCog-Bench is 560.

**CVR**    The original dataset link of CVR is github.com/serre-lab/CVR. It is under Apache License 2.0 (CVR LICENSE). CVR is an accepted paper by NeurIPS 2022 Datasets and Benchmarks track, so all of its data is free to use by public. We follow the same data generation pipeline in CVR to generate 309 samples.

**MaRs-VQA**    The image data of MaRs-VQA is from MaRs-IB (Chierchia et al., 2019) and annotated with context option by our team. It contains 18 questionnaires, each of questionnaire contains 80 matrix reasoning questions. The human study of MaRs-IB is rigorous. In MaRs-IB's original user study, all participants provided informed consent and all procedures were approved by UCL's ethical committee.

The paper and study results are under MIT License. All questionnaires are under Attribution-NonCommercial 3.0 (MaRs-IB LICENSE), which means it allows people to use the work, or adaptations of the work, for noncommercial purposes only, and only as long as they give credit to the creator. Thus, the MaRs-VQA dataset will under the same license.

After we download all questionnaires from MaRs-IB, we use two Python scripts to merge all question-option pairs from different questionnaires into the same sample set. Then, we generate Option Set A, Option Set B in Figure 6 by manipulating the size and image position of option images. After that, we annotate the language description of 4 options in 10 samples from the raw data. The language description is used as system prompt to guide GPT-4o to generate all description for all data in MaRs-VQA. Then, human annotators review the annotation and revise them. Finally, we publish all annotations as Option Set A, Option Set B, and Option Set C for MaRs-VQA. Figure 6 shows an example of each type of option.

The sub-task statistics of MaRs-VQA is in Table.

Compared to other zero-shot matrix reasoning dataset (Table 1) to evaluate matrix reasoning for MLLMs, MaRs-VQA has advantages list below:

- MaRs-VQA comprises 1,440 image instances designed by psychologists, making it the largest dataset for zero-shot matrix reasoning evaluation.

- MaRs-VQA includes a diverse range of data, such as variations in color, geometry, positional relationships, and counting.

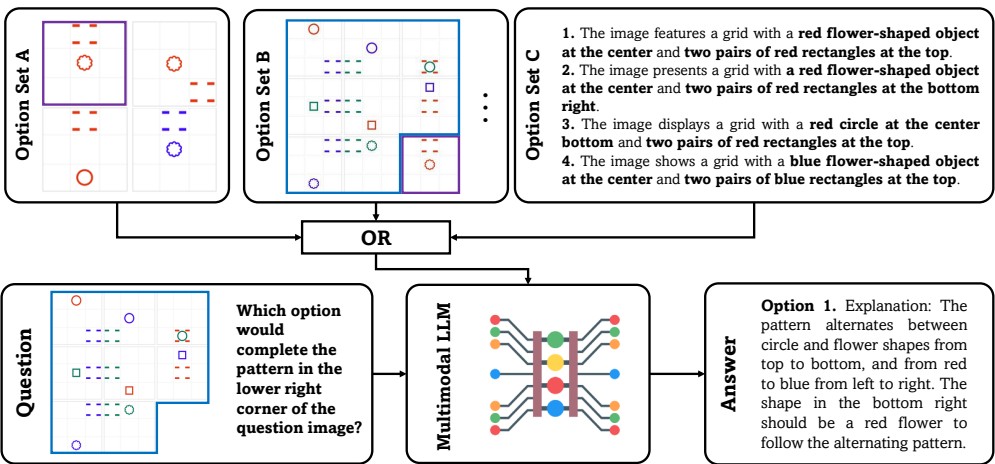

Figure 6: VQA Design of MaRs-VQA to evaluate Multimodal LLMs. The input set contains an image with a corresponding question and three sets of four-option images/contexts. Option Set A includes single-object images that can be filled into the blank region. Option Set B includes full 3x3 images containing all objects. Option C includes language descriptions for each option.

- The data source for MaRs-VQA is MaRs-IB (Chierchia et al., 2019), which is based on rigorous human studies. This dataset is widely recognized in the psychology community and has inspired numerous follow-up studies in child psychology and pediatrics. This is the first time we introduce it to the AI/ML community.

## C  EXPERIMENTAL SETTINGS

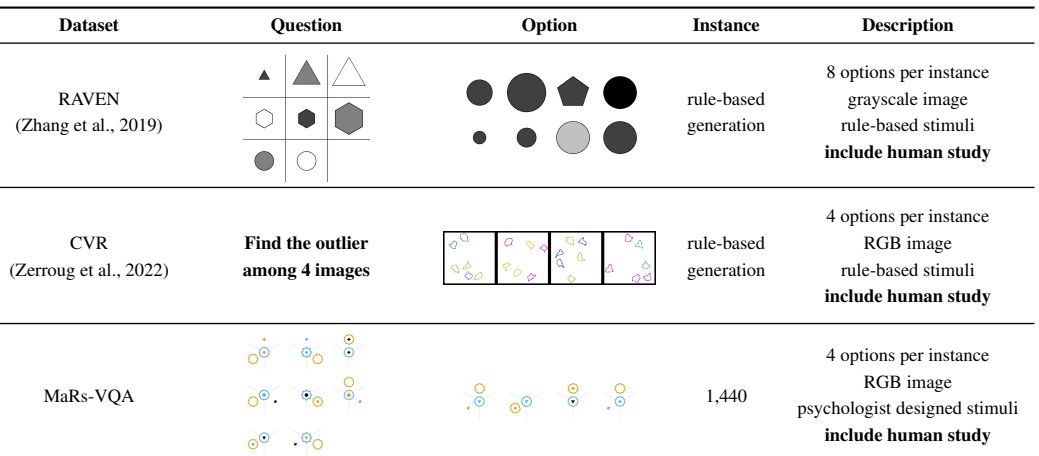

| Dataset | Question | Option | Instance | Description |
|---|---|---|---|---|
| RAVEN (Zhang et al., 2019) | | | rule-based generation | 8 options per instance
grayscale image
rule-based stimuli
**include human study** |
| CVR (Zerroug et al., 2022) | **Find the outlier among 4 images** | | rule-based generation | 4 options per instance
RGB image
rule-based stimuli
**include human study** |
| MaRs-VQA | | | 1,440 | 4 options per instance
RGB image
psychologist designed stimuli
**include human study** |

Table 7: Datasets in the VCog-Bench. Both the RAVEN and CVR are rule-based generated datasets. The test samples in MaRs-VQA are designed by psychologists from MaRs-IB.

### C.1  IMPLEMENTATION DETAILS

We used langchain to implement all closed-source MLLMs. The temperature of all models are 0 and the max token length is 1024. For all datasets, we follow their default image size, type settings for closed-source MLLMs. All experiments are run with three different random seeds, however, since we set temperature to 0, the final accuracy is the same for all random seeds.

For open-source models, we use the public available weights and data loader settings from the HuggingFace. InstructBLIP (Dai et al., 2024) and MiniGPT-4 (Zhu et al., 2023) are used their original GitHub repo to implement the zero-shot matrix reasoning inference pipeline. Testing is conducted using two NVIDIA RTX 4090 GPUs for 7B-sized VLMs and eight NVIDIA A100 80GB GPUs for VLMs larger than 7B. All experiments are run with three different random seeds, and the results are averaged.

## C.2 DIFFICULTY LEVELS IN MARS-VQA

| Difficulty Level | Question | Option | Description |
|---|---|---|---|
| 1 | | | Shape + Size |
| 2 | | | Color + Multi-object |
| 3 | | | Shape + Color + Position |
| 4 | | | Shape + Color + Multi-object |
| 5 | | | Shape + Color + Position + Multi-object |

Table 8: Explanation of Difficulty Levels.

Based on Figure 8, here is the explanation of difficulty levels presented in our paper:

- **Difficulty Level 1**: Single sub-task and two simple sub-tasks Description: The task involves only one changing attribute across the matrix reasoning—either shape, color, size, position, or multi-object. Or two simple attributes: (shape & color), (shape & size), (shape & position), (color & size), (color & position), (size & position). Example: Figure 4 (top-left) is a matrix reasoning task where only the size and color of the objects changes. This is a difficulty level 1 task.

- **Difficulty Level 2**: Two sub-tasks involving multi-object sub-task Description: The task involves multiple objects combined with one other changing attribute. The sub-task combinations are (multi-object & shape), (multi-object & color), (multi-object & size), (multi-object & position).

- **Difficulty Level 3**: Three simple sub-tasks combined Description: The task involves three changing attributes simultaneously. The sub-task combinations are (shape & color & size), (shape & position & size), (shape & position & color), (size & position & color).

- **Difficulty Level 4**: Three sub-tasks involving multi-object sub-task Description: The task involves multiple objects combined with two other changing attributes. The sub-task combinations are (multi-object & shape & color), (multi-object & shape & size), (multi-object & shape & position), (multi-object & color & position), (multi-object & color & size), (multi-object & position & size).

- **Difficulty Level 5 and Above**: Four or more Sub-tasks Description: The task involves combinations of four or five attributes. Example: Figure 4 (top-right) is a matrix reasoning task (shape & position & color & multi-objects) and its difficulty level is > 4.

As more attributes change simultaneously, the task becomes more complex, requiring higher levels of abstract reasoning to identify patterns. In addition, each additional changing element adds to the cognitive load, making it more challenging to discern the correct answer.

### C.3 MORE QUALITATIVE ANALYSIS

In this section, we further analyze the failure cases of GPT-4o. Correct reasoning is highlighted in green, while incorrect reasoning is marked in red. Although GPT-4o is sometimes able to extract a subset of key information from the question image, it frequently fails to arrive at the correct final answer. This is primarily due to critical features being either overlooked or inadequately utilized in the decision-making process. As a result, the final answers are often incorrect or only partially aligned with the relevant attributes. It reveals that visual working memory will be a key part to optimize the MLLM's performance in matrix reasoning problem.

### C.4 SYSTEM PROMPTS

For each dataset, we prepare custom system prompt. Their pipeline is similar. First, we created a system message prompt (see Figure 8, 9 for zero-shot inference, and Figure 10, 11, 12 for CoT) to guide the MLLM understanding the basic information of matrix reasoning tasks and the structure of the input, and formulating multiple-option images or contexts. The difference for zero-shot and CoT is we provide the guideline to encourage the model think the problem step-by-step based on extracting all useful information from structure $K = \{[r, a, o, s] | r \in \mathcal{R}, a \in \mathcal{A}, o \in \mathcal{O}, s \in \mathcal{S}\}$. The output format is a json structure including "Answer" and "Explanation" as keys.

## D FURTHER DISCUSSION ON LIMITATIONS AND FUTURE WORK

**Insights**  Unlike other VQA benchmarks, our work approaches the perspective of human visual cognition—an underexplored domain. Based on our experimental results, we offer the following insights for vision researchers:

- While scaling laws have some applicability to visual cognition tasks, merely increasing model size and training data is insufficient to achieve human-level performance.
- To demonstrate that VLMs possess strong visual cognitive abilities, it is crucial to evaluate them on zero-shot inference tasks like matrix reasoning—tasks characterized by simple visual content but requiring complex reasoning to find the correct answer.
- Unlike other multi-image visual reasoning benchmarks, VCog-Bench effectively highlights the performance gap between MLLMs and human cognition in these tasks.

From our main and ablation experiments, we observed that as task difficulty increases, the performance of MLLMs in multi-image reasoning scenarios deteriorates. Interestingly, providing language-based descriptions of each option (i.e., inputting the model with a single question image and context-based options) improved the models' performance compared to using multi-image options. This suggests that language still plays a significant role in the visual reasoning processes of current MLLMs and VLMs.

In contrast, human visual cognition—especially in children—allows individuals to solve matrix reasoning tasks without relying on advanced language reasoning capabilities. Children can often solve these tasks effectively by utilizing their visual working memory and pattern recognition skills.

One potential reason for the performance gap is that current MLLMs/VLMs may underemphasize the visual encoder relative to the language encoder. In many recently released VLMs, the visual module is much smaller than the language model module, and the visual encoders are frozen during Large Language Model (LLM) and alignment layer fine-tuning in open-sourced VLMs. This imbalance might limit the models' capacity to retain and process complex visual information during reasoning tasks.

To better retain visual information during the reasoning process, MLLMs may require more capable visual modules that can handle complex visual patterns and maintain this information throughout the reasoning steps. Moreover, optimizing the training process with end-to-end multimodal training—without freezing any layers in the visual modules—can be beneficial. Recent models have begun to explore end-to-end VLM fine-tuning, demonstrating the potential of this approach, though challenges remain such as the need for multi-round alignment. In the future, developing more advanced methods to effectively integrate visual and linguistic features will be crucial.

**Limitations** In the main paper, we briefly discussed the limitations of our work. Here, we provide a more in-depth discussion. First, our dataset is composed of limited publicly available matrix reasoning datasets, which must include human study results. The RAVEN and CVR datasets, created by the AI/ML community, were not developed following rigorous psychological research norms. Consequently, our benchmarking results, which utilize these datasets, should not be used to derive psychological or clinical conclusions. While MaRs-VQA addresses this problem, its samples cannot represent all formats of matrix reasoning found in IQ tests such as the WISC and the Cattell Culture Fair Intelligence Test (Cattell & Cattell, 1960). We cannot use these IQ tests directly because they are not freely available, and copyright restrictions usually prevent these pen-and-paper tasks from being adapted into computerized formats.

Second, the size of the datasets in VCog-Bench is relatively small compared with typical computer vision datasets, due to the inherent challenges involved in collecting matrix reasoning data. However, as we have argued in our paper, matrix reasoning should not be presented in typical machine learning settings—fine-tuning models on training sets and evaluating performance on test sets. Benchmarking MLLMs' visual reasoning performance should be conducted in a zero-shot inference setting, ensuring that all data in the test set are not included in the models' training data. Even compared with other recently released human-designed matrix reasoning datasets, ours is still the largest (see Table 1).

**Future Work** Although LLMs have achieved remarkable success in language understanding and generation, a significant portion of their parameters is dedicated to encoding linguistic patterns and memorizing factual information, which offers limited benefits for tasks requiring visual cognition. This disparity between Multimodal LLMs and humans indicates that merely increasing model size is insufficient to achieve human-level zero-shot inference in these domains. While our benchmark and baseline models represent a significant initial step, further data collection and in-depth human studies remain essential.

From our experimental results, we observe that current MLLMs have enhanced basic matrix reasoning capabilities, with models like GPT-4o and Gemini Pro 1.5 achieving significantly higher accuracy than random guessing across all three matrix reasoning tasks. By using Monte Carlo Tree Search to optimize the results via multi-round reasoning and exclusion, GPT-4o can achieve much better outcomes, albeit at the cost of increased inference time. We anticipate that the next generation of MLLMs will approach human-level performance in matrix reasoning. It is crucial to maintain these visual cognition-based benchmarks, continuously monitor the performance of newly released MLLMs, and encourage open-source MLLMs and VLMs to include matrix reasoning tasks for performance comparison.

Finally, we pose the open-ended question of whether MLLMs need to achieve or surpass human-level zero-shot inference capability in matrix reasoning tasks. Addressing this issue requires drawing on theories from cognitive science and psychology to understand the nature of human and MLLM intelligence. Matrix reasoning ability develops early in human neurodevelopment, with children as young as four providing sensible answers to simple matrix reasoning questions without additional training, making it a critical component of IQ tests. In contrast, LLMs and MLLMs rely on training data, fundamentally differing from how children develop cognitive abilities. However, we believe that these two learning processes share commonalities: both involve the gradual accumulation of skills and the ability to generalize from past experiences. Exploring these parallels can provide valuable insights into designing MLLMs that more closely mimic human visual cognition, ultimately leading to more advanced and capable models. Additionally, we observe that current open-source models achieve matrix reasoning performance very close to that of closed-source models. However, VLMs face challenges in supporting multiple images as input and managing visual memory. Addressing these challenges is a crucial direction for building more robust open-source VLMs in the future.

# E  ETHICS DISCUSSION

This research aims to advance LLMs and VLMs by providing a new benchmark for evaluating AI capabilities in visual reasoning. MaRs-VQA builds on the MaRs-IB (Attribution-NonCommercial 3.0 License), and VCog-Bench builds on MaRs-VQA, RAVEN (GPL-3.0 License), CVR (Apache License 2.0). All code and data are available on GitHub. No conflicts of interest exist among the study's contributors. More discussion on the ethical aspects of VCog-Bench is included in the Appendix. The annotation process is IRB approved by a clinical institute.

## E.1  NEGATIVE SOCIETAL IMPACTS

We foresee no direct negative societal impacts from our matrix reasoning benchmark. However, it could be misunderstood or misinterpreted as comparing AI "thought" to human cognition or misused to evaluate human abilities across demographics or ethnicity. We strongly caution against such misuse, as our datasets are not validated for human assessment.

Another concern relates to the future conclusion from our benchmark. While matrix reasoning is a crucial test for evaluating human intelligence, observing that VLMs with large model weights perform better on matrix reasoning tasks does not imply that the intelligence of MLLMs follows the same "scaling law" from the general domain. A comprehensive intelligence test requires accurate assessment using human-based tools, of which matrix reasoning is only one critical component. We cannot conclude that larger MLLMs can achieve human intelligence.

Additionally, there is a potential concern for discrimination against certain groups based on race, gender, or age in human study results. Although all human results in our experiment tables are sourced from previously published papers, we cannot guarantee that all previous research adhered to strict standards ensuring the inclusion of all groups in the human investigation process.

## E.2  MITIGATING BIAS AND NEGATIVE SOCIETAL IMPACTS

While the use of VCog-Bench and MaRs-VQA come with potential negative social impacts, there are viable mitigations that can address these concerns. These include adding instructions for proper use and restricting unethical human investigations. Users must be aware of the ethical implications associated with our benchmark and take appropriate measures to ensure its safe and responsible utilization.

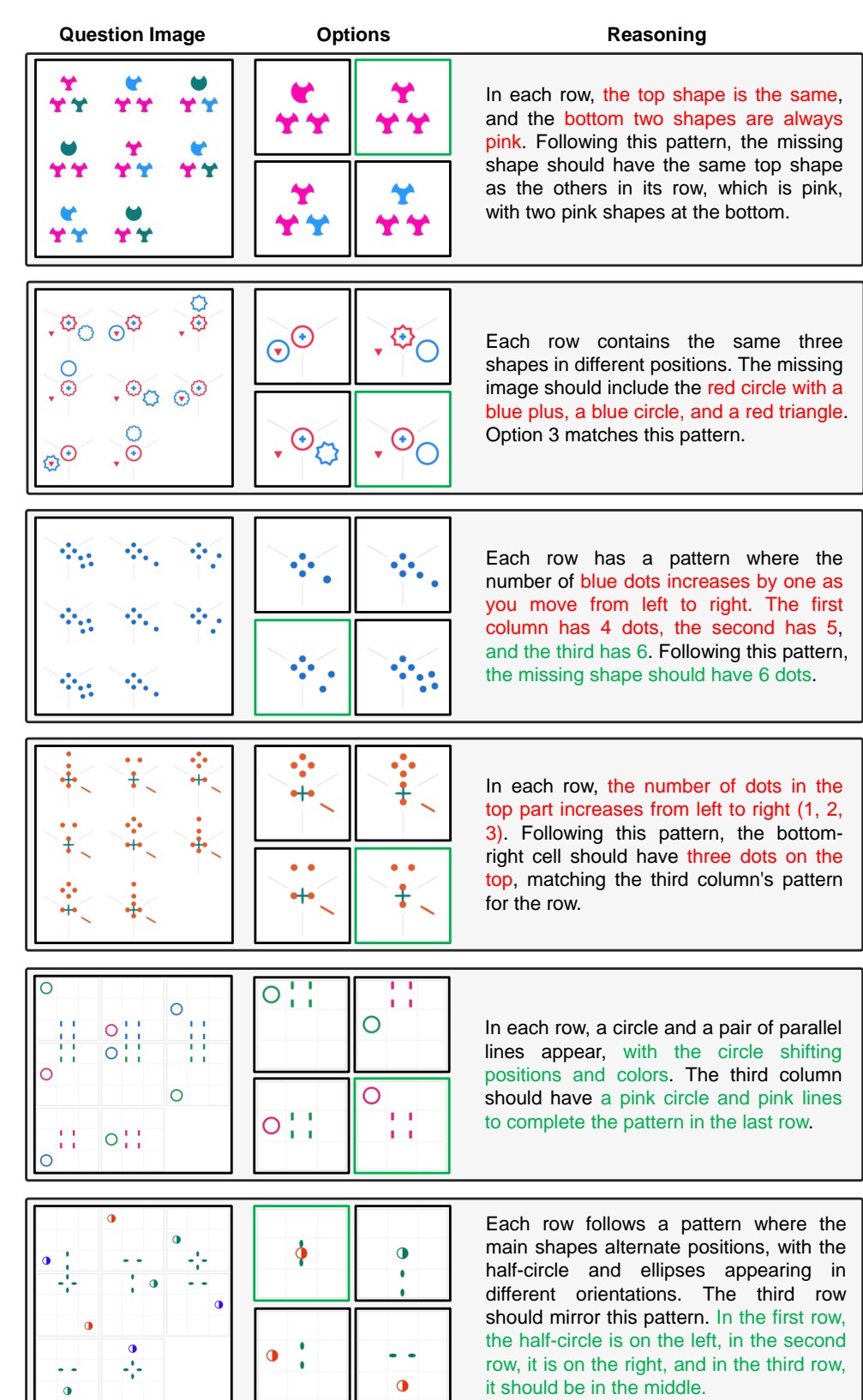

Figure 7: More visualization results for GPT-4o's reasoning.

**System Prompt for zero-shot inference for MaRs-VQA**

**System Message**

You are a helpful visual reasoning assistant that can solve abstract reasoning problems. Each task consisted of a question image with a 3 times 3 matrix. Eight of the nine resulting cells contained an abstract shape, while one cell on the bottom right-hand side of the matrix was empty. Your task is to complete the matrix by finding the missing shape among four possible alternatives. One of the option images is the correct answer. To select the correct missing shape, you have to deduce relationships between the shapes of the matrix. These shape characteristics varied along these dimensions: shape, color, size, and position in the matrix. You should only respond in the format as described below:

**Response Format**

*Answer:* The index of the correct answer, as a single letter.

Figure 8: System prompts for zero-shot MLLM inference of MaRs-VQA.

**System Prompt for zero-shot inference for RAVEN**

**System Message**

You are a helpful visual reasoning assistant solve abstract reasoning problem. Each task consisted of a question image with a 3 times 3 matrix. Eight of the nine resulting cells contained an abstract shape, while one cell on the bottom right-hand side of the matrix was empty. Your task is to complete the matrix by finding the missing shape among eight possible alternatives. One of the option image is the correct answer. To select the correct missing shape, you have to deduce relationships between the shapes of the matrix. These shape characteristics varied along five dimensions: number, shape (triangle, square, pentagon, hexagon, circle), color (five colors from white to black), size (five size from small to large) and position in the matrix.

You should only respond in the format as described below:

**Response Format**

*Answer:* The index of the correct answer, as a single letter.

Figure 9: System prompts for zero-shot MLLM inference of RAVEN.

---

**System Prompt for MLLMs with CoT for MaRs-VQA**

**System Message**

You are a helpful visual reasoning assistant that can solve abstract visual reasoning problems. Each task consisted of a question image with a 3 times 3 matrix. Eight of the nine resulting cells contained an abstract shape, while one cell on the bottom right-hand side of the matrix was empty. Your task is to complete the matrix by finding the missing shape among four possible alternatives. One of the options is the correct answer.

The first step is to describe what is the attribute and relationship between each attribute in each cell of the 3 times 3 question image. The attributes can be number, position, shape, size, and color. The cell may contain multiple attributes. The relation might be '3 times 3 sub-blocks', 'rotation', 'insideness'.

The second step is to summarize the relation of three patterns in the first row of the question image, the relation of three patterns in the second row of the question image, the relation of two patterns in the third row of the question image.

Answer this question: What are the row-based high-order rules in the question image?

Based on the description for each option, answer this question: What is the constraint of all options?

Finally, infer what are the potential attributes, objects, relations in the missing cell?

You should only respond in the format as described below:

**Response Format**

*Explanation:* The step-by-step reasoning for the answer.
*Answer:* The index of the correct answer, as a single letter.

Figure 10: System prompts for MLLM CoT inference of MaRs-VQA.

---

**System Prompt for MLLMs with CoT for RAVEN**

**System Message**

You are a helpful visual reasoning assistant that can solve abstract visual reasoning problems. Each task consisted of a question image with a 3 times 3 matrix. Eight of the nine resulting cells contained an abstract shape, while one cell on the bottom right-hand side of the matrix was empty. Your task is to complete the matrix by finding the missing shape among eight possible alternatives. One of the option images is the correct answer.

The first step is to summarize the relation of three patterns in the first row of the question image, the relation of three patterns in the second row of the question image, the relation of two patterns in the third row of the question image. What is this relation? The features in the patterns can be constant, progression, arithmetic, distribute three. Try to describe this relationship.

The second step is to describe what is the attribute and relationship between each attribute in each cell of the 3 times 3 cells question image and four option images. The attributes can be number; shape (triangle, square, pentagon, hexagon, circle); colour (five colors: white, light gray, gray, dark gray, black); size (five size: tiny, small, medium, large, huge); and positional relation (inside outside relation, left right relation, top down relation, two times two sub-blocks, 3 times 3 sub-blocks). The cell may contain multiple attributes.

Finally, give me the answer based on step 1-2.

You should only respond in the format as described below:

**Response Format**

*Explanation:* The step-by-step reasoning for the answer.
*Answer:* The index of the correct answer, as a single letter.

Figure 11: System prompts for CoT MLLM inference of RAVEN.

**System Prompt for MLLMs with CoT for CVR**

**System Message**

You are a helpful visual reasoning assistant that can solve abstract reasoning problems. Each task consisted of four option images. Your task is to identify which image is different from the other three? To find the correct answer, you have to deduce relationships inside each image and then find the difference.

The first step is to describe what is the attribute and relationship between each attribute in four option images. The features can be number of objects; shape; color; size, relationship of colour or shape or size or direction among objects; and positional relation of objects (inside outside relation, left right relation, top bottom relation, adjacent relation). Each image may contain multiple attributes and multiple relations.

Based on the description and image for each option, answer this question: What is the constraint / similarity of most of the options?

Finally, infer which image is the outlier?

You should only respond in the format as described below:

**Response Format**

*Explanation:* The step-by-step reasoning for the answer.
*Answer:* The index of the correct answer, as a single letter.

Figure 12: System prompts for CoT MLLM inference of CVR.

