# OpenReview forum: "Benchmarking Visual Cognition of Multimodal LLMs via Matrix Reasoning"
_ICLR.cc/2025/Conference — ICLR 2025 Conference Withdrawn Submission_

### Official Review · Reviewer_ssS6 · 2024-11-03

**Soundness:** 2
**Presentation:** 3
**Contribution:** 3
**Rating:** 6
**Confidence:** 3

**Summary:**

This paper introduces a new dataset called MaRs-VQA and a benchmark named VCog-Bench to evaluate how well MLLMs understand and reason about visual information. It focuses on testing these models using matrix reasoning tasks, which are inspired by established intelligence tests like RPM and the WISC. These tasks require advanced visual reasoning, which is still challenging for current AI models. The paper compares how different MLLMs perform in these tasks compared to humans, showing a clear gap in capabilities. The contributions include making MaRs-VQA, the largest dataset designed by psychologists for matrix reasoning, and introducing VCog-Bench as a new standard for evaluating visual cognition. The experiments show the current limitations of MLLMs in dealing with abstract visual problems and provide suggestions for future improvements.

**Strengths:**

- The paper focuses on a challenging area of visual cognition relevant for assessing human-like intelligence. Unlike many studies that emphasize perception-based tasks (e.g., object detection), this work addresses higher-level reasoning and working memory. The MaRs-VQA dataset, based on established psychological tests, offers a diverse and validated dataset for visual cognition.
- The work highlights a gap between human and machine cognition in abstract visual reasoning, even for advanced models like GPT-4o. By focusing on more sophisticated visual cognition, the paper encourages the development of models that can reason about images abstractly, not just recognize them. This has important implications for the progress of AGI.
- The paper is well-structured, with clear definitions of each dataset, model, and experimental setup. The comparison between human and model performance is effective in showcasing current limitations and areas for improvement.

**Weaknesses:**

- The paper does not explore how changes in the visual complexity of MaRs-VQA impact model performance. It is unclear whether models are sensitive to subtle changes (e.g., color gradient variations or object overlaps), which could provide important insights into model robustness and areas for improvement.
- One of the weaknesses is the lack of detailed analysis into why specific models fail at particular reasoning tasks. However, considering that the primary goal of the paper is to propose a new dataset, this focus on benchmarking rather than in-depth analysis of is understandable.

**Questions:**

- Could the authors explore how changes in the visual complexity of MaRs-VQA impact model performance? Are models sensitive to subtle changes, such as color gradient variations or object overlaps?
- The paper showed zero-shot performance as an evaluation criterion. Could few-shot learning approaches be explored as a transitional step to understand whether MLLMs can improve their performance incrementally?
- How might MLLMs be modified to better retain visual information during the reasoning process? Want to hear the authors' opinion.

**Details Of Ethics Concerns:**

There do not appear to be any significant ethical concerns with this work.

---

> ### Author Response · Authors · 2024-11-24
> **Response to Reviewer ssS6 (Part 1)**
>
> **Visual complexity of MaRs-VQA impact model performance**
>
> Thank you for your insightful suggestions. We have conducted experiments on various subtle changes (from Appendix Table 5). The experimental results indicate that GPT-4o with CoT reasoning performs relatively well in distinguishing object sizes. However, for the other four sub-tasks, its performance remains almost unchanged. This is likely because object size is the simplest sub-task in matrix reasoning, accounting for only 16% of the dataset, with most samples classified as difficulty levels 1–2. In contrast, the multi-object sub-task is the most challenging for GPT-4o with CoT. This is because multi-object sub-tasks often appear in conjunction with one or two additional sub-tasks within the same VQA sample. We appreciate your suggestion to explore fine-grained VQA experiments, and we will incorporate all relevant findings into the paper.
>
> While we are eager to explore other subtle changes, such as color gradient variations or object overlaps as you proposed, these cases are currently very limited in the MaRs-VQA dataset. To enable further study, we plan to expand the dataset and benchmark by adding more images that include these variations.
>
> | Visual complexity | Accuracy |
> | --- |  --- |
> | Shape |  33.96  |
> | Color |  35.72  |
> | Size |  63.26  |
> | Position |  31.70  |
> | Multi-Object |  31.48  |
>
>
> **Few-shot learning**
>
> Thank you for your valuable suggestions. We have conducted experiments incorporating selected few-shot examples to enhance the performance of multi-image CoT reasoning. The results show that using 1-shot and 3-shot question-option-answer pairs increased the accuracy on MaRs-VQA from 34% to 36% gradually. However, when we extended the number of examples to 5, the performance did not improve further. Additionally, The maximum token of GPT-4o restricts the inclusion of more few-shot examples. These findings suggest that while few-shot examples can help the model better grasp the matrix reasoning problem, they do not significantly improve the visual reasoning capabilities of the MLLM for matrix reasoning tasks. We appreciate your suggestion to explore few-shot experiments, and all relevant information, including the difficulty level of each few-shot sample will be incorporated into the paper.
>
> Additionally, we conducted experiments to improve baseline performance using a multi-round CoT approach.
>
> The GPT-4 multi-round CoT is an extension of the multi-image CoT based on the Tree of Thoughts (ToT) framework. It utilizes Monte Carlo Tree Search to invoke GPT-4 multiple times to select the best option candidates. We observed that using multi-round tree search can improve the accuracy from approximately 34% to 42%, although it is significantly slower than the single-round CoT, with each inference case taking over 30 seconds in multi-round running.
>
> While adjusting the system prompts with few-shot learning and employing multi-round CoT inspired by ToT can enhance performance—albeit at the expense of time and computational resources—these methods still perform substantially worse than human participants.
>
> We will include these detailed results in the revised version of our manuscript:
>
> | Prompt | Accuracy |
> | --- |  --- |
> | GPT-4o 0-shot |  30.21  |
> | GPT-4o CoT |  33.96  |
> | GPT-4o CoT + 1-shot |  35.22  |
> | GPT-4o CoT + 3-shot |  36.10  |
> | GPT-4o CoT + 5-shot |  36.03  |
> | GPT-4o multi-round tree search CoT |  41.96  |
> | GPT-4o multi-round tree search CoT + 1-shot |  42.08  |

---

> ### Author Response · Authors · 2024-11-24
> **Response to Reviewer ssS6 (Part 2)**
>
> **How to let better retain visual information during the reasoning process**
>
> Thank you for your insightful question.
>
> Based on our observations from the ablation experiments in Table 4, we found that as the difficulty of the tasks increases, the performance of MLLMs deteriorates in multi-image reasoning scenarios. Interestingly, we also observed that providing language-based descriptions of each option (input the model with single question image and context-based options) improved the models' performance compared to using multi-image options. This suggests that language still plays a significant role in the visual reasoning processes of current MLLMs and VLMs.
>
> In contrast, human visual cognition—especially in children—allows individuals to solve matrix reasoning tasks without relying on advanced language reasoning capabilities. Children can often solve these tasks effectively by utilizing their visual working memory and pattern recognition skills.
>
> One potential reason for the performance gap is that current MLLMs / VLMs may underemphasize the visual encoder relative to the language encoder. In many recently released VLMs, the visual module is much smaller than the language model module and the visual encoders are frozen during LLM and alignment layer finetuning in open-sourced VLMs. This imbalance might limit the models' capacity to retain and process complex visual information during reasoning tasks.
>
> To better retain visual information during the reasoning process, MLLMs may need more capable visual modules that can handle complex visual patterns and maintain this information throughout the reasoning steps. In addition, the training process can be optimized with end-to-end multimodal training without freezing any layers in the visual modules. Llama-3.2 shows a possibility of end-to-end VLM finetuning, but it still needs multi-round alignment.
> Another aspect is applying the theory of predictive coding to incorporate multiple visual encoders into the design of VLMs. We are working on the VLM optimization and will continuously update it.
>
> We appreciate your interest in our work and believe that these directions hold promise for advancing the field. We are committed to investigating these avenues further and look forward to sharing our findings with the research community.

---

### Official Review · Reviewer_ksNJ · 2024-11-03

**Soundness:** 2
**Presentation:** 2
**Contribution:** 2
**Rating:** 5
**Confidence:** 3

**Summary:**

This paper introduces MaRs-VQA as a new matrix reasoning VQA dataset, and VCog-Bench as a visual cognition benchmark to evaluate the matrix reasoning performance of 16 existing MLLMs in the zero-shot setting. The thorough experiments qualitatively reveal the visual cognition gap between MLLMs and humans in matrix reasoning problems.

**Strengths:**

1. The proposed MaRs-VQA dataset and VCog-Bench benchmark help establish the cognitative training and evaluation pipeline for multi-modal large language model.

2. The evaluations experiments are comprehensive and include extenstive MLLMs.

3. Some of the insights found in experiments can inspire more future investigations.

**Weaknesses:**

1. The motivation is unclear. It is not clear how the proposed MaRs-VQA differs from the privious ones. From Table 1, it seems the most remarkable difference is the introduction of RGB image.

2. The authors claim that "This setting makes current matrix reasoning assessment an ill-posed problem because such tests accurately reflect reasoning capability only when subjects engage without prior training, i.e., in zero-shot inference settings." It is quite confusing since the training-testing paradigm is the common methodology. As long as the training and test sets do not overlap, why does this pattern not make sense?

3. In multi-image reasoning evaluation of Table 2, only GPT and Claude evaluated. Why the open-source models not included?

4. In Table 4, the authors claim that "The difficulty level is based on the complexity of color, size, geometry, positional relationships, and object counting." However, it is not clear how the difficulty level is related to the mentioned elements. More details should be presented.

**Questions:**

Please refer to the weakness part.

---

> ### Author Response · Authors · 2024-11-24
> **Response to Reviewer ksNJ (Part 1)**
>
> **Motivation**
>
> We appreciate the opportunity to clarify the motivation and distinct contributions of our MaRs-VQA dataset. MaRs-VQA differs from previous datasets in several significant ways:
>
> | Motivation | Contribution |
> | --- |  --- |
> | Largest Dataset | MaRs-VQA is the largest among all matrix reasoning datasets under the same setting, offering a more extensive resource for evaluating MLLMs and VLMs in matrix reasoning tasks. |
> | Unique Annotations | It is the only dataset that includes comprehensive vision-language annotations, which are essential for assessing models' understanding of both visual content and corresponding linguistic descriptions. |
> | Validated by Psychometrics Experts | All matrix reasoning tasks in MaRs-VQA have been validated by psychometrics researchers, ensuring psychological validity. |
> | Human Performance Data | The inclusion of human study records allows for direct comparisons between model performance and human reasoning abilities. |
> | IRB Approval | We obtained IRB approval from a clinical institute, underscoring the ethical and scientific rigor of our dataset and potential chance to complete the studies on comparing MLLM's matrix reasoning capability with kids. |
> | Public Availability | Unlike other datasets such as kosmos-iq50 (NeurIPS 2023) and the Visual Reasoning Benchmark (COLM 2024), which are broken links or not currently open-sourced, MaRs-VQA is available via an anonymous link. This open access promotes transparency and facilitates further research. |
>
> While the introduction of RGB images is one visible difference, the key contributions of MaRs-VQA lie in its size, detailed annotations, psychological validation, and accessibility. These features collectively make it a valuable and novel resource for advancing research in visual cognition and AI.
>
> **About training-testing paradigm**
>
> Thank you for your insightful question. The key point is that matrix reasoning tests aim to measure innate reasoning capabilities and the ability to identify patterns and relationships in novel situations. If we were to train models specifically on matrix reasoning tasks, we would be providing them with specialized experience that humans do not have when taking these tests. This could lead to an overestimation of the models' reasoning abilities, as they might be leveraging learned patterns specific to the training data rather than demonstrating genuine generalization and reasoning skills.
>
> In human psychometrics, matrix reasoning tests like Raven's Progressive Matrices and the Wechsler Intelligence Scale for Children are designed to assess visual reasoning abilities without prior specific training on similar tasks. Children taking these tests typically do not receive any specialized training in matrix reasoning beforehand. Instead, they rely on their general cognitive skills developed through everyday experiences in natural scenes. Remarkably, most children can perform well on the matrix reasoning sections even without prior exposure to similar tasks [1].
>
> Therefore, we argue that evaluating MLLMs and VLMs on matrix reasoning tasks in a zero-shot setting, without prior training on similar tasks, more closely mirrors the conditions under which humans are assessed in visual cognition. This approach allows us to better evaluate a model's inherent reasoning abilities and its capacity to generalize from previous experiences, rather than its ability to learn from specific examples of the task.
>
> Our motivation stems from interdisciplinary insights, including contributions from a co-author with over 30 years of experience in psychometrics, human cognition, and autism research. By aligning our evaluation methodology with the principles of human cognitive assessments, we aim to develop a visual cognition benchmark that challenges models in a manner analogous to how humans are challenged in matrix reasoning tasks.
>
> References:
> [1] Laurence, P. G., & Macedo, E. C. (2023). Cognitive strategies in matrix-reasoning tasks: State of the art. Psychonomic Bulletin & Review, 30(1), 147-159.

---

> ### Author Response · Authors · 2024-11-24
> **Response to Reviewer ksNJ (Part 2)**
>
> **Open-source models not included**
>
> Thank you for your question. We appreciate the opportunity to clarify our experimental choices.
>
> Before conducting the experiments reported in Table 2, we extensively tested the performance of various closed-source and open-source models, including the latest VLMs such as InternVL-2-76B and Qwen2-VL-72B. We observed that all these open-source models faced significant challenges with the multi-image reasoning tasks:
>
> - Same Responses: The open-source models tended to generate the same answer (often the same option letter like 'A') across all tasks.
>
> - Lack of Step-by-Step Reasoning: These models were unable to produce the detailed, step-by-step reasoning that is essential for solving matrix reasoning problems. This contrasts with the capabilities of models like GPT-4 and Claude, which could articulate their reasoning process. Similarly, we evaluated the performance of the closed-source MLLM Gemini-Pro-1.5 on the multi-image reasoning tasks. However, it frequently (over 30% of cases) responded with statements like "I can't answer this question," which made it difficult to include its results alongside those of GPT-4o and Claude 3 (Example in Figure 4).
>
> Due to these limitations, including the open-source models in Table 2 would not have provided valuable insights. Our goal in Table 2 was to present a clear and informative comparison of models capable of engaging with multi-image reasoning at a level that allows for meaningful analysis.
>
> To address this issue and provide a more inclusive evaluation, we designed a second task within the VCog-Bench, as presented in Table 3. This task uses one 3x3 question image from matrix reasoning along with context-only based options, which are more accessible to both open-source and closed-source models.
>
> We will update our manuscript to clarify these points, ensuring that our methodology and the rationale behind our experimental design are transparent.
>
>
> **Difficulty level in Table 4**
>
> Thank you for your question. We'd like to clarify how the difficulty levels in Table 4 are determined and how they relate to variations in color, size, geometry (shape), positional relationships, and if there are multiple objects presented in the task.
>
> Difficulty Levels Explained:
>
> - Difficulty Level 1: Single sub-task and two simple sub-tasks
> Description: The task involves only one changing attribute across the matrix reasoning—either shape, color, size, position, or multi-object. Or two simple attributes: (shape & color), (shape & size), (shape & position), (color & size), (color & position), (size & position).
> Example: Figure 4 (top-left) is a matrix reasoning task where only the size and color of the objects changes. This is a difficulty level 1 task.
>
> - Difficulty Level 2: Two sub-tasks involving multi-object sub-task
> Description: The task involves multiple objects combined with one other changing attribute. The sub-task combinations are (multi-object & shape), (multi-object & color), (multi-object & size), (multi-object & position).
>
> - Difficulty Level 3: Three simple sub-tasks combined
> Description: The task involves three changing attributes simultaneously. The sub-task combinations are (shape & color & size), (shape & position & size), (shape & position & color), (size & position & color).
>
> - Difficulty Level 4: Three sub-tasks involving multi-object sub-task
> Description: The task involves multiple objects combined with two other changing attributes. The sub-task combinations are (multi-object & shape & color), (multi-object & shape & size), (multi-object & shape & position), (multi-object & color & position), (multi-object & color & size), (multi-object & position & size).
>
> - Difficulty Level 5 and Above: Four or more Sub-tasks
> Description: The task involves combinations of four or five attributes.
> Example: Figure 4 (top-right) is a matrix reasoning task (shape & position & color & multi-objects) and its difficulty level is > 4.
>
> How Elements Relate to Difficulty:
> Increasing Complexity: As more attributes change simultaneously, the task becomes more complex, requiring higher levels of abstract reasoning to identify patterns.
> Cognitive Load: Each additional changing element adds to the cognitive load, making it more challenging to discern the correct answer.
> We will update the manuscript to include these detailed explanations and provide illustrative examples for each difficulty level. This should make the relationship between difficulty levels and the task elements clear.
>
> Thank you again for your valuable feedback. Your input helps us enhance the clarity of our work.

---

### Official Review · Reviewer_rmqC · 2024-11-04

**Soundness:** 2
**Presentation:** 2
**Contribution:** 2
**Rating:** 5
**Confidence:** 5

**Summary:**

This paper introduces a new dataset MaRs-VQA and a new benchmark VCog-Bench to evaluate the zero-shot visual cognition capability of MLLMs with a matrix reasoning task. It requires MLLMs to discern relationships among patterns in a set of images and extrapolate to predict subsequent patterns. This paper proposes two evaluation pipelines of the proposed VCog-Bench: (1) Multi-image reasoning via CoT and (2) direct image input and text output. The evaluations are performed on various MLLMs of both APIs and open-source models.

**Strengths:**

1. The paper is clear and easy to follow.

2. The evaluations are performed on various MLLMs of both APIs and open-source models.

3. It reveals the current MLLMs still need to improve on matrix reasoning.

**Weaknesses:**

1. The contribution of this paper is incremental.
- The proposed MaRs-VQA and VCog-Bench are all sourced from existing well-built datasets, including MaRs-IB, RAVEN, and CVR.
The proposed multi-image reasoning via CoT method is an application of CoT to a particular task. It is not a general solution for other tasks.
- The conclusion of the limitation of current MLLMs is not supported by sufficient evidence and is not convincing. How the author attains the conclusion that current MLLMs have Limited Use of Visual Information and Restricted Visual Working Memory needs to be clarified.
2. The experiments show that the MLLMs perform much worse than humans. It is unknown if it is because the MLLMs do not understand the task to perform. The author may evaluate MLLMs with in-context learning, which can take one or two QA paris as examples.
3. Additionally, it will be interesting to discuss if the MLLMs can easily attain the ability to solve matrix reasoning via training in a small number of cases. For example, the model can be trained on a small subset of MaRs-VQA and evaluated on VCog-Bench.

**Questions:**

Due to the incremental contributions of this paper, I tend to be borderline negative in the current stage. Please refer to the weakness section for detailed comments.



###################

Thank you for your explanation; it addresses part of my concerns. However, I still feel that the contribution of this work falls below the standard expected for ICLR. Firstly, MaRs-VQA and VCog-Bench are derived from the questionnaires of MaRs-IB, RAVEN, and CVR. While these datasets structure the task as VQA, I believe the questionnaire images themselves inherently serve as both questions and answers. As such, MaRs-VQA and VCog-Bench merely reformulate them, which represents a relatively minor contribution compared to the original questionnaires. Secondly, the discussion on the limitations of current MLLMs emphasizes their difficulties in handling multiple sub-tasks simultaneously. However, it is just a conjecture, as the evidence provided in the paper is insufficient to substantiate this claim.

Overall, I will keep my rating.

---

> ### Author Response · Authors · 2024-11-24
> **Response to Reviewer rmqC (Part 1)**
>
> **MaRs-VQA and VCog-Bench are all sourced from existing well-built datasets**
>
> Thanks for your question. We would like to clarify the distinctions and contributions of our work. Firstly, MaRs is not a dataset. While MaRs provides source images from questionnaires designed for human assessments, it does not constitute a dataset suitable for evaluating AI models, because it does not have VQA annotations. In designing MaRs-VQA, we created such annotations, including each option in the VQA, and question description. Our process is common in building VQA datasets, where existing images are re-annotated to create novel challenges for VLMs (e.g., ScienceQA, which repurposed images and questions from elementary and high school science curricula; DriveLM, which used images from nuScenes).
>
> | | MaRs-VQA |  MaRs |
> | --- |  --- | --- |
> | Description |  Dataset  | Questionnaire |
> | Data Type | Image & Text | Image |
>
> The other matrix reasoning datasets in Table 1 are also sourced from some questionnaires (see column Source in Table 1).
>
> VCog-Bench vs. RAVEN and CVR:
>
> Similarly, while we sourced visual content from RAVEN and CVR, the original datasets do not include any language-based options or VQA annotations suitable for MLLMs and VLMs evaluation tasks.
>
> In VCog-Bench, we extensively re-annotated RAVEN and CVR by introducing VQA, presenting the tasks in a completely different way. This transformation creates a unique benchmark that assesses models' visual cognition abilities in a manner not previously explored in RAVEN and its follow-up research.
>
> **Multi-image CoT in our task**
>
> Thank you for your thoughtful feedback.
>
> We want to test if the MLLMs can have a better performance by CoT like humans in matrix reasoning tasks. We don't aim at proposing a novel method for entirely solving the problem.
>
> Both the Multi-image CoT (section 4.1) and the VLM pipeline (section 4.2) are integral parts of our benchmark design to adapt to different input types. They are existing methods but we re-design the structure to let them adapt to the matrix reasoning tasks. Our benchmarks and baseline methods are designed to serve as a first step to solve this problem and inspire further research, also providing tools for evaluating and improving the reasoning abilities of MLLM models in visual cognition. We are considering organizing challenges on MaRs-VQA and VCog-Bench to further engage the ML and cognition community.
>
> **Limitation of current MLLMs**
>
> Thank you for your insightful comment. In our ablation experiments, we observed that as the difficulty of the tasks increases, the performance of MLLMs deteriorates in multi-image reasoning. The difficulty level in our tasks is defined by the number of sub-tasks involved—specifically, variations in color, size, geometry (shape), positional relationships, and the presence of multiple objects.
>
> This observation suggests that MLLMs are affected by the cognitive load associated with processing multiple sub-tasks simultaneously, which is closely related to the concept of visual working memory. For instance, in Figure 4 (top-left) of our paper, GPT-4o demonstrates strong performance by accurately explaining its solution. It effectively captures the visual differences in shape and color within the 3x3 question matrix and applies this information to select the best-fitting option image.
>
> However, for more challenging tasks that involve a higher number of sub-tasks (the others in Figure 4), GPT-4o tends to extract only a small portion of the key information from the question image. This limited extraction means that critical features are either overlooked or not effectively utilized in selecting the correct option. Consequently, the final answers are often incorrect or only partially related to the relevant attributes.
>
> This decline in performance with increased task difficulty highlights a limitation in current MLLMs' ability to handle complex visual reasoning tasks that require integrating multiple visual attributes simultaneously. It underscores the need for improvements in models' visual working memory capabilities to more closely mimic human cognition in processing and reasoning about multifaceted visual information.

---

> ### Author Response · Authors · 2024-11-24
> **Response to Reviewer rmqC (Part 2)**
>
> **MLLMs do not understand the task to perform & In-context learning**
>
> Thank you for your valuable suggestions. We have conducted experiments incorporating selected few-shot examples to enhance the performance of multi-image CoT reasoning. The results show that using 1-shot and 3-shot question-option-answer pairs increased the accuracy on MaRs-VQA from 34% to 36% gradually. However, when we extended the number of examples to 5, the performance did not improve further. Additionally, The maximum token of GPT-4o restricts the inclusion of more few-shot examples. These findings suggest that while few-shot in-context learning can help the model better grasp the matrix reasoning problem, they do not significantly improve the visual reasoning capabilities of the MLLM for matrix reasoning tasks. We appreciate your suggestion to explore few-shot experiments, and all relevant information, including the difficulty level of each few-shot sample will be incorporated into the paper.
>
>
> | Method | Accuracy |
> | --- |  --- |
> | GPT-4o 0-shot |  30.21  |
> | GPT-4o CoT |  33.96  |
> | GPT-4o CoT + 1-shot |  35.22  |
> | GPT-4o CoT + 3-shot |  36.10  |
> | GPT-4o CoT + 5-shot |  36.03  |
>
>
> **About training in a small number of cases**
>
> Thank you for your insightful suggestion. We agree that it would be interesting to explore whether MLLMs can attain the ability to solve matrix reasoning tasks through visual instruction tuning / RLHF finetuning on a small number of cases. However, in the context of our current work, training on the benchmark data is intentionally not allowed. (line 048-050 in the original paper)
>
> Our primary objective with VCog-Bench is to provide a platform to evaluate the zero-shot visual cognition abilities of MLLMs and VLMs, assessing how well these models can generalize to unfamiliar tasks without prior exposure. This zero-shot setting is designed to mimic human-like reasoning in novel situations, highlighting the inherent capabilities of the models rather than their ability to learn from specific training data. In human psychometrics, matrix reasoning tests like Raven's Progressive Matrices and the Wechsler Intelligence Scale for Children are designed to assess visual reasoning abilities without prior specific training on similar tasks. Children taking these tests typically do not receive any specialized training in matrix reasoning beforehand. Instead, they rely on their general cognitive skills developed through everyday experiences in natural scenes. Remarkably, most children can perform well on the matrix reasoning sections even without prior exposure to similar tasks [1].  (line 489-508 in the original paper)
>
> Allowing models to train on a subset of MaRs-VQA and then evaluating them on VCog-Bench would change the nature of our benchmark from zero-shot inference to training-testing paradigm, which falls outside the scope of our current study. Our benchmark is the first zero-shot paradigm to compare human’s visual cognition with MLLMs and VLMs.
>
> We will update our manuscript to discuss the dataset license to explicitly state that training on the benchmark data is not permitted. This clarification will help ensure that the benchmark is used as intended and that comparisons between models remain consistent. We appreciate your suggestion helping us completing the definition of the benchmark.
>
>
> References:
> [1] Laurence, P. G., & Macedo, E. C. (2023). Cognitive strategies in matrix-reasoning tasks: State of the art. Psychonomic Bulletin & Review, 30(1), 147-159.

---

### Official Review · Reviewer_yD7a · 2024-11-05

**Soundness:** 3
**Presentation:** 2
**Contribution:** 1
**Rating:** 3
**Confidence:** 4

**Summary:**

This paper studies the high-level visual cognition abilities that require multi-image reasoning, by investigating matrix reasoning. A new benchmark is proposed:  MARS-VQA which contains 1440 instances and VCogBench to evaluate matrix reasoning abilities.  The paper finds that similar to previous findings with RAVEN's matrices and other similar tests, state of the art MLLMs struggle at matrix reasoning and perform worse or slightly better than random (25%) performance on a four-way classification task.

**Strengths:**

1. Paper is well structured and experiments are comprehensive in terms of the number of models evaluated.
2. MLLM evaluation is an important challenge and the paper seeks to address that question with a connection to human psychometric evaluation.

**Weaknesses:**

1. The claim that matrix reasoning has been "proven to be used to test human intelligence" or that "matrix reasoning is an important reflection of many fundamental capabilities of human intelligence" are to say the least, as controversial as saying "IQ Tests" are a true reflection of human intelligence.
2. The benchmark is restricted to shapes but could have potentially also used natural images. In my opinion, making claims about human visual cognition where the test data is purely symbolic is an overclaim. It could be an evaluation of human symbolic cognition.
3. The experiments include prompts designed for this task -- the influence of this choice of prompts on the performance is unclear.

**Questions:**

1. What are the insights from the paper? Why are we interested in testing matrix reasoning of VLMs and why do we want VLMs to succeed at this task? Besides observing that VLMs aren't good at this task, what should VLM researchers and developers should take away from this work?

---

> ### Author Response · Authors · 2024-11-24
> **Response to Reviewer yD7a (Part 1)**
>
> **Relationship between matrix reasoning and human intelligence**
>
> Thank you for bringing up this important point. We acknowledge that matrix reasoning does not fully encapsulate all aspects of human intelligence. However, it remains a valuable tool for measuring symbolic understanding capabilities, which are essential components of visual cognition abilities in general intelligence and pave the way toward Artificial General Intelligence (AGI) [1].
>
> Traditional VQA benchmarks, such as ScienceQA, DriveLM, VizWiz VQA, LLaVA-Med focus on testing the ability of the MLLM in answering questions based on domain-specific knowledge. These benchmarks evaluate the task-specific skills of such models, which correspond to the bottom level of the cognitive ability hierarchy proposed in [1]'s Figure 1. In contrast, matrix reasoning tasks differ from traditional VQA problems because they require higher-order cognitive processes, such as abstract reasoning, pattern recognition, and analogical thinking. These tasks demand that models interpret and manipulate symbolic representations, whereas previous VQA problems often focus on direct recognition or retrieval of information from visual content. By engaging these advanced cognitive abilities, matrix reasoning aligns more closely with human cognition, corresponding to the second level of "Broad Cognitive Abilities" in [1]'s Figure 1. The term "broad abilities" is often used in opposition to "local generalization," emphasizing the importance of assessing a model's capacity to generalize across a wide range of tasks and domains.
>
> Our work represents one of the first attempts to measure how close MLLMs are to achieving visual cognition ability of AGI by evaluating their performance on matrix reasoning tasks. We believe this approach can inspire future studies in this field, encouraging the development of models that better mimic human-like intelligence and reasoning abilities.
>
> We will revise our manuscript to clarify these points and ensure that we accurately represent the scope and intent of our work without overstating the implications regarding general intelligence.
>
> Reference:
> [1] Chollet, F. (2019). On the measure of intelligence. arXiv preprint arXiv:1911.01547.
>
>
> **Why not natural images**
>
> Thank you for your insightful feedback. We understand your concern regarding the use of purely symbolic data in our benchmark and the implications for making claims about human visual cognition.
>
> We chose to focus on matrix reasoning tasks because they provide a well-defined specific cognitive processes related to visual abstraction and reasoning. Natural images have complex backgrounds, shapes and colors, which makes it hard to assess the ability of models in understanding each single element. In contrast, symbolic data disentangles these components and has a clean background. We are able to control the variable in each question to make a detailed study [1,2]. While matrix reasoning is symbolic in nature, it taps into fundamental aspects of visual cognition essential for complex visual understanding.
>
> In addition, matrix reasoning tests aim to measure innate reasoning capabilities and the ability to identify patterns and symbolic relationships in novel situations. Using a natural image cannot help us to navigate the failures VLMs make quantitatively, due to the confounding variables appearing in it. We observed different results in matrix reasoning tasks: even when we increase the size of MLLMs and VLMs, there remains a significant performance gap compared to human abilities.
>
> References:
> [1] Laurence, P. G., & Macedo, E. C. (2023). Cognitive strategies in matrix-reasoning tasks: State of the art. Psychonomic Bulletin & Review, 30(1), 147-159.
> [2] Lovett, A., & Forbus, K. (2017). Modeling visual problem solving as analogical reasoning. Psychological review, 124(1), 60.

---

> ### Author Response · Authors · 2024-11-24
> **Response to Reviewer yD7a (Part 2)**
>
> **Influence of prompts design**
>
> Thank you for your valuable suggestions regarding our experiments. We have indeed adjusted the CoT strategy used in Table 2 multiple times to optimize performance. The GPT-4 CoT approach we selected combines option summaries, question image row summaries, and step-by-step reasoning, as illustrated in Figure 3 of our paper.
>
> Here are the results of our experiments using different system prompt designs:
>
> | Prompt | Accuracy |
> | --- |  --- |
> | GPT-4o 0-shot |  30.21  |
> | GPT-4o 0-shot+step-by-step reasoning |  26.47  |
> | GPT-4o 0-shot+option summary |  31.69  |
> | GPT-4o 0-shot+ question image row summary |  30.88  |
> | GPT-4o CoT |  33.96  |
>
> As a follow-up study to improve the baseline, we also compared the CoT strategy with CoT augmented by few-shot learning samples. In the few-shot in-context learning setup, we provided GPT-4 with example question-option-answer pairs as part of the input. Additionally, we conducted experiments to improve baseline performance using a multi-round CoT approach.
>
> The GPT-4 multi-round CoT is an extension of the multi-image CoT based on the Tree of Thoughts (ToT) framework. It utilizes Monte Carlo Tree Search to invoke GPT-4 multiple times to select the best option candidates. We observed that using multi-round tree search can improve the accuracy from approximately 34% to 42%, although it is significantly slower than the single-round CoT, with each inference case taking over 30 seconds in multi-round running.
>
> While adjusting the system prompts with few-shot learning and employing multi-round CoT inspired by ToT can enhance performance—albeit at the expense of time and computational resources—these methods still perform substantially worse than human participants.
>
> We will include these detailed results in the revised version of our manuscript:
>
> | Prompt | Accuracy |
> | --- |  --- |
> | GPT-4o 0-shot |  30.21  |
> | GPT-4o CoT |  33.96  |
> | GPT-4o CoT + 1-shot |  35.22  |
> | GPT-4o CoT + 3-shot |  36.10  |
> | GPT-4o CoT + 5-shot |  36.03  |
> | GPT-4o multi-round tree search CoT |  41.96  |
> | GPT-4o multi-round tree search CoT + 1-shot |  42.08  |
>
> **Insights from our paper**
>
> Compared to other VLM benchmarks, our work begins from the perspective of human visual cognition—an underexplored domain proposed by [1]. Based on our experimental results, we offer the following insights for vision researchers:
>
> - Scaling laws apply to visual cognition tasks in a limited scope; however, simply increasing model size and training data does not achieve human-level performance. (line 474-477 in the original paper)
> - To demonstrate that VLMs possess strong visual cognitive abilities, it is crucial to evaluate them on zero-shot inference tasks like matrix reasoning—tasks characterized by simple visual content but need complex reasoning to answer the question. (line 474-477 in the appendix)
> - Unlike other multi-image visual reasoning benchmarks, VCog-Bench effectively highlights the performance gap between Multimodal Large Language Models (MLLMs) and human cognition in such tasks. (line 489-508 in the original paper)
>
> These insights underscore the need for developing new approaches beyond scaling to bridge the gap between current models and human visual cognition. We hope our benchmark encourages further exploration in this important but underserved area.
>
> Reference:
> [1] Chollet, F. (2019). On the measure of intelligence. arXiv preprint arXiv:1911.01547.
>
> We appreciate your interest in our work and believe that these directions hold promise for both the cognition and ML community. We are committed to investigating these avenues further and look forward to sharing our findings with the research community.

---

> ### Comment · Reviewer_yD7a · 2024-12-01
>
> Hi authors,
>
> Thank you for your detailed response.
> - For weakness 1, I appreciate the clarification about the motivation being "AGI" evaluation motivated by the book by Chollet.
> - For weakness 2, my concern remains that this could be a test for symbolic cognition, but the paper claims this to be visual cognition.
> - Thanks for the prompt-related experiments to address weakness 3.
> - for the question, thanks for providing these additional insights.
>
> I will raise my score to 4, but there is much scope for improving this paper. I cannot recommend acceptance at this stage.

---

> > ### Author Response · Authors · 2024-12-02
> > **Response to Reviewer yD7a (Part 3)**
> >
> > Dear Reviewer yD7a,
> >
> > Thank you for your thoughtful review and for taking the time to engage with our work.
> >
> > We appreciate your acknowledgment of our clarification regarding the motivation behind our "AGI" evaluation of weakness 1.
> >
> > Regarding your concern about weakness 2, we understand that the distinction between symbolic cognition and visual cognition in our test may not have been sufficiently clear to ML and AI researchers outside the neuroscience & cognitive science community. Based on your suggestion, we would like to follow the naming idea from KiVA, another concurrent work submitted to ICLR for focusing on another small aspect (visual analogies) of visual cognition, we would like to rename our work to "Benchmarking Symbolic Visual Cognition of Multimodal LLMs via Matrix Reasoning". I think this can solve your concerns. In addition, the concept of "matrix Reasoning" has already included the scope of the paper, focusing on the matrix reasoning cognition test, but not natural images.
> >
> > We are glad that the prompt-related experiments we conducted to address weakness 3 were helpful. Your suggestions have significantly contributed to improving our work.
> >
> > Thank you again for your constructive feedback and for raising your score.
> >
> > Reference:
> > KiVA: https://openreview.net/forum?id=vNATZfmY6R

---

### Author Response · Authors · 2024-11-24
**Rebuttal by Authors**

Dear Reviewers,

We sincerely appreciate your thorough and insightful comments on our paper. Your feedback has been invaluable in helping us clarify our contributions and strengthen our work. We have carefully considered each of your points and have also conducted additional experiments to address your concerns. Please find our detailed responses below. Here is a summary of our motivation to help you recap the contribution of our paper:

The foundation of our research lies in matrix reasoning tasks similar to those found in the visual cognition part in intelligence tests, such as the Wechsler Intelligence Scale for Children (WISC) and Raven's Progressive Matrices (RPM). Assessing matrix reasoning essentially involves evaluating a subject's reasoning and learning potential in real-time, within an unfamiliar environment. For instance, when presented with an unfamiliar matrix reasoning task, like those in MaRs-VQA, both human and AI subjects must identify high-level patterns (i.e., abstract concepts) to solve the puzzle. Thus, the training-testing paradigm can not be adopted into this benchmark directly.

François Chollet articulated a similar perspective in his book On the Measure of Intelligence (see Figure 1 in his book). For AI in general domains, such as Multimodal LLMs, it is crucial to measure broad abilities—potentially generality itself—rather than specific skills. Our benchmark paper aims to encourage the ML / Cognition community to reconsider this cognition challenge in the era of LLMs and explore solutions for designing the next generation of Multimodal LLMs that can adapt to unfamiliar visual reasoning tasks, much like children do.

We sincerely welcome any further feedback you may have. We deeply appreciate the time and effort you have already dedicated to reviewing our work. We would be grateful if you could review our responses once more and let us know if they fully address your concerns.

Best regards,
Authors of Submission2479

---

> ### Author Response · Authors · 2024-11-30
> **Official Comment by Authors**
>
> Dear Reviewers,
>
> We have uploaded an updated version of our paper. In this revision, we have:
>
> - Modified the Abstract and Introduction to avoid making controversial claims, as suggested by Reviewer yD7a.
> - Highlighted why this task is zero-shot inference rather than a training-testing paradigm, addressing the concerns of Reviewers rmqC and ksNJ.
> - Concluded with additional insights in the Discussion section and the Appendix, as suggested by Reviewer yD7a and Reviewer ssS6.
> - Included new experiments and detailed information about the dataset, such as difficulty levels and discussions on visual working memory, in both the main text and the Appendix.
> - Redrawn figures to incorporate more discussion on prompt selection.
>
> As the discussion period draws to a close, we sincerely welcome any further feedback you may have. We deeply appreciate the time and effort you have dedicated to reviewing our work. We would be grateful if you could review our responses once more and let us know if they fully or partially address your concerns, and whether our revisions are moving in the right direction. Please feel free to share any additional questions or comments you may have about the paper. We are committed to continually improving our work, and your feedback is invaluable to us.
>
> Best regards,
> The Authors

---

### Note · Authors · 2025-01-22

I have read and agree with the venue's withdrawal policy on behalf of myself and my co-authors.